# FACTS: A Factored State-Space Framework For World Modelling

**Li Nanbo**[1*]**, Firas Laakom**[1]**, Yucheng Xu**[2]**, Wenyi Wang**[1]**, Jürgen Schmidhuber**[1,3]

[1]Center of Excellence for Generative AI, KAUST, Saudi Arabia
[2]School of Informatics, University of Edinburgh, United Kingdom
[3]The Swiss AI Lab, IDSIA, USI & SUPSI, Switzerland

## ABSTRACT

World modelling is essential for understanding and predicting the dynamics of complex systems by learning both spatial and temporal dependencies. However, current frameworks, such as Transformers and selective state-space models like Mambas, exhibit limitations in efficiently encoding spatial and temporal structures, particularly in scenarios requiring long-term high-dimensional sequence modelling. To address these issues, we propose a novel recurrent framework, the **FACT**ored **S**tate-space (**FACTS**) model, for spatial-temporal world modelling. The FACTS framework constructs a graph-structured memory with a routing mechanism that learns permutable memory representations, ensuring invariance to input permutations while adapting through selective state-space propagation. Furthermore, FACTS supports parallel computation of high-dimensional sequences. We empirically evaluate FACTS across diverse tasks, including multivariate time series forecasting, object-centric world modelling, and spatial-temporal graph prediction, demonstrating that it consistently outperforms or matches specialised state-of-the-art models, despite its general-purpose world modelling design. Code available at: ⌗ https://github.com/NanboLi/FACTS.

## 1 INTRODUCTION

World modelling (Schmidhuber, 1990b; 2015; Ha & Schmidhuber, 2018) aims to create an internal representation of the environment for an AI system, enabling it to represent (Hafner et al., 2019; 2023), understand (Schrittwieser et al., 2020; Hafner et al., 2020), and predict (Ha & Schmidhuber, 2018; Micheli et al., 2022) the dynamics of complex environments. This capability is crucial for various domains, including autonomous systems, robotics, and financial forecasting, where accurate predictions depend on effectively capturing both spatial and temporal dependencies (Hafner et al., 2019; Ha & Schmidhuber, 2018). Consequently, spatial-temporal learning (Liu et al., 2024; Hochreiter & Schmidhuber, 1997; Wu et al., 2023a; Oreshkin et al., 2019) emerges as a key challenge in world modelling, as approaches must balance the complexities of modelling high-dimensional sequential data while maintaining robust long-term predictive power.

Despite significant advancements, current spatial-temporal learning frameworks, used in world modelling, based on Transformers (Schmidhuber, 1992a; Vaswani et al., 2017; Schlag et al., 2021) and RNNs (Schmidhuber, 2015; Ha & Schmidhuber, 2018; Hafner et al., 2019; 2020) backbones, face limitations in fully capturing the complexities of high-dimensional spatial-temporal data. Transformer, though powerful (Chen et al., 2022; Robine et al., 2023; Micheli et al., 2022), are inefficient for long-term tasks due to their quadratic scaling and limited context windows (Zhang et al., 2022). On the other hand, RNNs provide a more structured approach to sequential data. However, their efficacy is hindered by the vanishing gradients (Hochreiter, 1991; Pascanu et al., 2013). The primary challenges in spatial-temporal learning arise from the high dimensionality of the data and the necessity to preserve long-term dependencies (Hochreiter et al., 2001; Tallec & Ollivier, 2018).

Recently, there has been a growing interest in Structured State-Space Models (SSMs) for world modelling (Gu & Dao, 2023; Hafner et al., 2023; Samsami et al., 2024) using latent state-space

---

[*]Correspondence to nanbo.li@kaust.edu.sa

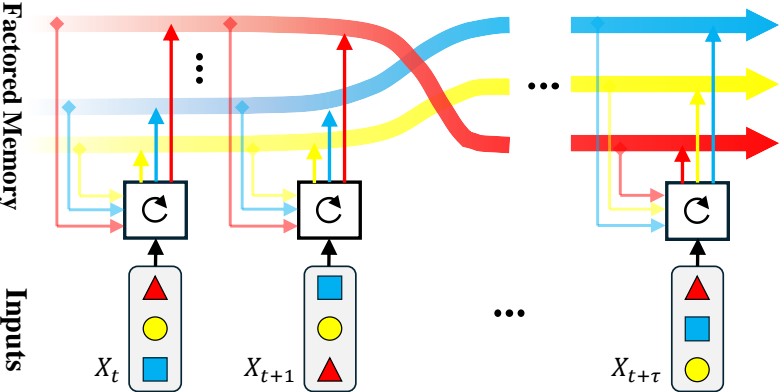

Figure 1: **Overview of FACTored State-space (FACTS) Architecture.** The FACTS framework constructs a factored state-space memory, allowing for flexible representations (e.g. graphs and sets). Sequential inputs (e.g. $X_t$) are processed through a selective memory-input interaction mechanism (denoted by the circular icon ↻), which determines how the inputs interact with and update factored memory. The different coloured pathways represent distinct latent factors, whose dynamics evolve over time based on these interactions. The design ensures that the memory update is permutation-invariant with respect to the input features, enabling FACTS to capture and track meaningful algorithmic regularities for accurate future predictions.

representations. These representations allow for the modelling of underlying dynamics, where latent states evolve over time according to the governing equations (Wang et al., 2024b). However, while SSMs have demonstrated improved capacity for capturing temporal dynamics (Gu & Dao, 2023; Wang et al., 2024b; Baron et al., 2023), they often lack efficient mechanisms to handle high-dimensional spatial data. To address this limitation, recent approaches often impose rigid structural constraints on their state spaces, such as diagonal (Gu & Dao, 2023; Gupta et al., 2022a;b) or block-diagonal structures (Dao & Gu, 2024), to capture invariant components throughout the sequence. This assumption, that specific dimensions of the state space correspond to consistent patterns over time, can be restrictive in world modelling scenarios where the relationship between state-space dimensions and input features evolves dynamically.

For example, in a dynamic system involving multiple agents (e.g. robots or sensors), where the positions of the agents change over time, to capture this dynamism, current SSMs require learning distinct representations for essentially identical scenarios at each time step as the agent locations change. This redundancy leads to an inefficient use of model capacity and data, ultimately limiting the model's ability to effectively capture the dynamics of the interactions between inputs and states. Therefore, there is a need for a consistent dynamic mapping between inputs and latent states to enhance spatial-temporal modelling capabilities and enable more efficient history compression (Schmidhuber, 1992b; 2003), which is essential for robust long-term prediction power. Another limitation of current SSMs is their inability to capture redundancy in the input space itself. In many cases, each agent's state (e.g., position, speed, direction) may contributes to the world's overall understanding, but the identity or order of the agents do not matter, i.e., only their interactions are crucial for making accurate predictions. In such instances, swapping agents should not alter the predictions and the world understanding. However, current SSM-based methods, typically based on linear transformations, fail to account for this and can perceive identical scenarios as different based on input order, hindering its ability to capture regularities and making them unsuitable for sequential modelling in various applications.

To address these challenges, in this paper, we propose the FACTS model, a novel recurrent framework for spatial-temporal world modelling. FACTS conceptualises the input as a set of nodes and introduces a permutable memory that can incorporate complex structures. Through selective memory-input routing, input features are dynamically assigned to distinct state-space factors, i.e., explanatory latent representations, that capture the underlying dynamics of the system. This formulation ensures input permutation invariance in the state-space memory, allowing the model to learn consistent factor representations over time, even when the spatial or temporal relationships between input features and factors change as illustrated in Figure 1. Additionally, by treating inputs as a set of

nodes, FACTS: i) can incorporate a wider range of input structures, e.g., images or graphs or sets. ii) maintains consistent representations of inputs (e.g., agents), regardless of their order. This allows the model to capture regularities in both spatial/local modalities and temporal dependencies, enhancing its memory efficiency and long-term prediction capabilities through more efficient history compression (Schmidhuber, 1992b; 2003). To validate our proposed world modelling approach, we conduct an extensive empirical analysis across multiple tasks, such as multivariate time series forecasting and object-centric world modelling, demonstrating that FACTS consistently matches or exceeds the performance of specialised state-of-the-art models. This confirms its robustness and versatility in addressing complex high-dimensional sequential tasks.

In summary, our main contributions are as follows:

- We introduce **FACT**ored **S**tate-space (FACTS), a novel recurrent framework that incorporates a permutable memory structure, enabling flexible and efficient modelling of complex spatial-temporal dependencies.
- FACTS dynamically assigns input features to distinct latent state-space factors, ensuring effective history compression and enhancing long-term prediction power.
- We formally and empirically show that FACTS achieves consistent factor representations over time, regardless of changes in the spatial order of input features over time, providing robustness in dynamically evolving environments.
- We validate the robustness and predictive power of FACTS through extensive world modelling experiments, demonstrating its superior or competitive performance across multivariate time-series forecasting, object-centric world modelling, and spatial-temporal graph prediction tasks.

## 2 PRELIMINARIES

Structured-state space Models (SSMs) have their roots in the classic Kalman filter (Kalman, 1960), where they process a $m$-dimensional input signal $\boldsymbol{x}(t) \in \mathbb{R}^m$ into a $d$-dimensional latent state $\boldsymbol{z}(t) \in \mathbb{R}^d$, which is then projected onto an output signal $\boldsymbol{y}(t) \in \mathbb{R}^n$. The general form of an SSM is expressed as follows:

$$\dot{\boldsymbol{z}}(t) = \boldsymbol{A}(t)\boldsymbol{z}(t) + \boldsymbol{B}(t)\boldsymbol{x}(t) \tag{1}$$

$$\boldsymbol{y}(t) = \boldsymbol{C}(t)\boldsymbol{z}(t) + \boldsymbol{D}(t)\boldsymbol{x}(t), \tag{2}$$

where $\dot{\boldsymbol{z}}(t) = \frac{d}{dt}\boldsymbol{z}(t)$ indicates the time derivative of the state. The matrices $\boldsymbol{A}(t) \in \mathbb{R}^{d \times d}$, $\boldsymbol{B}(t) \in \mathbb{R}^{d \times m}$, $\boldsymbol{C}(t) \in \mathbb{R}^{n \times d}$, and $\boldsymbol{D}(t) \in \mathbb{R}^{n \times m}$ present the state, input, output, and feed-forward matrices, respectively. In systems without direct feedthrough, $\boldsymbol{D}(t)$ becomes a zero matrix. Furthermore, since the original system operates in a continuous domain, discretisation is often used (Wang et al., 2024b; Smith et al., 2023), resulting in the general discrete-time formulation of SSM:

$$\boldsymbol{z}_t = \overline{\boldsymbol{A}}_t \boldsymbol{z}_{t-1} + \overline{\boldsymbol{B}}_t \boldsymbol{x}_t \tag{3}$$

$$\boldsymbol{y}_t = \boldsymbol{C}_t \boldsymbol{z}_t \tag{4}$$

with $\overline{\boldsymbol{A}}_t$, $\overline{\boldsymbol{B}}_t$, and $\boldsymbol{C}_t$ govern the dynamics driven by the input sequence $\boldsymbol{x}_{\leq t}$, with different constructions (Wang et al., 2024b; Gu & Dao, 2023; Dao & Gu, 2024) influencing the expressiveness and efficiency of the model. If we denote the state vector with $\boldsymbol{h}$, we can see that equation 3-equation 4 form is equivalent to the RNN dynamics. Hence, similarly to RNNs, the system, in equation 3-4, is inherently sequential, which limits parallel processing.

**Parallelisation and the selective mechanism** As shown in Blelloch (1990); Smith et al. (2023) if $\overline{\boldsymbol{B}}_t$ constructed independently of $\boldsymbol{z}_{t-1}$, the linear recurrence in equation 3 can be expanded as equation 5:

$$\boldsymbol{z}_t = \sum_{s=0}^{t} \bar{\boldsymbol{A}}_{t:s}^{\times} \bar{\boldsymbol{B}}_s \boldsymbol{x}_s, \tag{5}$$

where $\bar{\boldsymbol{A}}_{t:s}^{\times} := \bar{\boldsymbol{A}}_{t+1}...\bar{\boldsymbol{A}}_{s+2}\bar{\boldsymbol{A}}_{s+1}$; $\bar{\boldsymbol{A}}_{t+1} = \boldsymbol{I}$; $\bar{\boldsymbol{B}}_0 \boldsymbol{x}_0 = \boldsymbol{z}_0$ with respect to some initialisation. This expansion not only allows for parallel computation of the linear terms, but also reveals the direct connection established between distant inputs/observations along the sequential dimension, e.g. $\boldsymbol{x}_0$

and $\boldsymbol{x}_t$ with $t \gg 0$, thereby facilitating the capture of long-term dependencies. Furthermore, integrating a selective mechanism (Gu & Dao, 2023) by constructing $\bar{\boldsymbol{A}}_t$ and $\bar{\boldsymbol{B}}_t$ as functions of each input leads to the following formulation:

$$\boldsymbol{z}_t = \bar{\boldsymbol{A}}(\boldsymbol{x}_t)\boldsymbol{z}_{t-1} + \bar{\boldsymbol{B}}(\boldsymbol{x}_t)\boldsymbol{x}_t \tag{6}$$

$$= \sum_{s=0}^{t} \bar{\boldsymbol{A}}^{\times}(\boldsymbol{x}_{t:s})\bar{\boldsymbol{B}}(\boldsymbol{x}_s)\boldsymbol{x}_s \tag{7}$$

$$\boldsymbol{y}_t = \boldsymbol{C}(\boldsymbol{x}_t)\boldsymbol{z}_t \tag{8}$$

This formulation enables content-aware compression of the historical information, addressing the issue of memory decay in long-sequence modelling. Such selective mechanism is foundational to the effectiveness of modern state-space models (Gu & Dao, 2023; Dao & Gu, 2024). Additionally, as indicated in Eq. equation 6, SSMs can support parallel computation, since the term $\bar{\boldsymbol{B}}(\boldsymbol{x}_t)\boldsymbol{x}_t$ remains independent of the preceding state $\boldsymbol{z}_{t-1}$.

## 3 PROPOSED FRAMEWORK: FACTS

In this section, we introduce our proposed framework: **FACTS** (**FACT**ored **S**tate-space) model, a novel class of recurrent neural networks designed with a structured state-space memory. FACTS is characterised by two key features: **permutable state-space memory**, which allows for flexible representation of system dynamics with more complex structures and **invariant recurrence** with respect to permutations of the input features, ensuring consistent modelling of underlying factors in the world across different time steps.

One key intuition behind the "permutable state-space memory" in FACTS is the principle of *history compression* (Schmidhuber, 1992b; 2003), which emphasises the need to eliminate redundant information in sequence modelling while uncovering algorithmic regularities. This principle is essential for effective long-sequence modelling with high-dimensional data, as it improves generalisation by reducing the accumulation of unnecessary information. Existing SSMs often address this challenge by imposing fixed structural constraints on their state spaces, such as diagonal or block-diagonal structures, to capture invariant components that persist throughout the sequence (Gu & Dao, 2023; Gupta et al., 2022a;b; Dao & Gu, 2024). However, these fixed structural priors assume that specific dimensions of the state space correspond to consistent and specific factors over time. This assumption can be limiting in world-modelling scenarios where the relationship between state-space dimensions and input features evolves dynamically. For instance, in video sequence modelling, factors may correspond to moving objects, and the spatial location of these objects (i.e., pixel positions) changes from frame to frame. In such cases, the model needs to adapt to these changes, but current SSMs formulations struggle to maintain consistent factor representations due to their rigid structural constraints, i.e., in equation 7 the matrices $\bar{\boldsymbol{B}}(\boldsymbol{x}_t)$ must not only select relevant information for modelling sequence dynamics but also account for the changing relative orders between subspaces of $\boldsymbol{z}_{t-1}$ and $\boldsymbol{x}_t$, which can evolve over time. This introduces additional complexity, leading to noise and redundancy that can hinder effective history compression.

### 3.1 FACTS FORMULATION

The FACTS framework is formalised as a class of structured state-space model, which can capture the dynamic interactions between the latent factors and the input features. To facilitate this dynamic *factorisation*, i.e., the process of identifying and disentangling meaningful factors from the input data over time, at each time step $t$, we conceptualise the hidden state $Z_t$ as a graph and hence, the state-space memory is represented as a set of nodes that correspond to the latent factors. The input features $X_t$ are also treated as another set of nodes. Formally, let

$$Z_t = \{\boldsymbol{z}_t^1, \boldsymbol{z}_t^2, \ldots, \boldsymbol{z}_t^k\} \qquad\qquad X_t = \{\boldsymbol{x}_t^1, \boldsymbol{x}_t^2, \ldots, \boldsymbol{x}_t^m\} \tag{9}$$

where $Z_t$ denote the set of $k$ latent factors at time step $t$ and $m$ is the number of input features. By formulating both sets as nodes, FACTS is inherently invariant to permutations of both input features and factors. Then, to efficiently learn the optimal connections between these two sets, we propose a graph-based routing mechanism that can effectively match input features with the corresponding factors .i.e., learn edges between nodes in $Z_t$ and $X_t$, reflecting the strength of each correspondence between each input feature and each latent factor.

**Dynamic selective state-space updates** In analogy with the standard SSM dynamics equation 3-equation 4, the evolution of the latent factors of FACTS is governed by the following modified state-space dynamics:

$$Z_t = \bar{\boldsymbol{A}}_t \odot Z_{t-1} + \bar{\boldsymbol{B}}_t \odot \boldsymbol{U}_t \tag{10}$$

$$\boldsymbol{y}_t = Dec(\boldsymbol{C}_t \odot Z_t) \tag{11}$$

Here, $Z_t$ represents the state-space memory at time $t$, which stores the latent factors. The terms $\bar{\boldsymbol{A}}_t$, $\bar{\boldsymbol{B}}_t$, and $\boldsymbol{C}_t$ are selective state-space model parameters responsible for controlling the information flow between the previous memory $Z_{t-1}$ and the input features $X_t$. The symbol $\odot$ denotes element-wise multiplication, while $Dec$ is a permutation-invariant decoder applied to the latent factors.

Compared to the standard SSM dynamics, i.e., equation 3-equation 4, we note two key differences: (i) FACTS relies on element-wise multiplication, instead of matrix multiplication, to conserve the invariance properties. (ii) $x_t$ in equation 3 is replaced with $\boldsymbol{U}_t = (Z_{t-1}, X_t)$, which is a key element in FACTS that models the interactions between the memory $Z_{t-1}$ and the input features $X_t$.

**The attention-based router** Before diving into the details of the different parts of equation 10 and equation 11, we first introduce the routing mechanism used in this work. To maintain the recurrent permutability of equation 10, the routing mechanism between memory and inputs must dynamically assign input features to consistent factors. This can be done using an attention-based routing mechanism defined as follows:

$$Z_{t-1} \; \circlearrowright_{\phi,\psi,\varphi} \; X_t = \text{softmax}\left(\frac{\phi(Z_{t-1})\psi^T(X_t)}{\sqrt{d}}\right)\varphi(X_t) \tag{12}$$

where the operator $\circlearrowright$ learns the relationships between the memory $Z_{t-1}$ and input features $X_t$, dynamically determining which features correspond to which latent factors. The functions $\phi$, $\psi$, and $\varphi$ represent the query, key, and value mappings, respectively, and are applied row-wise to the memory and input features.

**Factorisation process** The term $\boldsymbol{U}_t = \boldsymbol{U}(Z_{t-1}, X_t)$ in equation 10 is crucial for capturing the interactions between the memory and the input features. Note that in prior works (Gu & Dao, 2023; Dao & Gu, 2024) $\boldsymbol{U}_t$ is typically constructed as function of the current input $X_t$ only. In this paper, we argue that, similar to the gating in RNNs vs LSTMs (Hochreiter & Schmidhuber, 1997), it is more effective use both $X_t$ and $Z_{t-1}$ to conserve long term dependencies. This interaction plays a key role in factorisation, which refers to the process of binding the input features to specific memory items, effectively uncovering the underlying factors. In the FACTS framework, the memory at the previous time step $Z_{t-1}$ serves as the prior over the latent factors, and $\boldsymbol{U}_t$ is the factor momentum that guides the evolution of these factors across time. This factor momentum is computed as:

$$\boldsymbol{U}_t = Z_{t-1} \; \circlearrowright_{\phi_U,\psi_U,\varphi_U} \; X_t \tag{13}$$

where $\phi_U$, $\psi_U$, and $\varphi_U$ are its corresponding query, key, and value mappings.

**Selectivity through memory-input routing** The selective state-space model parameters $\bar{\boldsymbol{A}}_t$, $\bar{\boldsymbol{B}}_t$, and $\boldsymbol{C}_t$ are constructed through interactions between the memory and the input features, ensuring that both the memory and the inputs jointly decide which information should be retained or updated. These parameters are computed as follows:

$$\Delta_t = Z_{t-1} \; \circlearrowright_{\phi_\Delta,\psi_\Delta,\varphi_\Delta} \; X_t \qquad\qquad \bar{\boldsymbol{A}}_t = \exp(\alpha\Delta_t) \tag{14}$$

$$\bar{\boldsymbol{B}}_t = \Delta_t \odot (Z_{t-1} \; \circlearrowright_{\phi_B,\psi_B,\varphi_B} \; X_t) \qquad\qquad \boldsymbol{C}_t = Z_{t-1} \; \circlearrowright_{\phi_C,\psi_C,\varphi_C} \; X_t \tag{15}$$

Here, $\Delta_t$ is a step size introduced for discretisation, and the functions $\phi_\Delta$, $\psi_\Delta$, and $\varphi_\Delta$ (as well as their counterparts for $\bar{\boldsymbol{B}}_t$ and $\boldsymbol{C}_t$) are responsible for mapping the memory and inputs to their respective selective parameters. The exponential function $\exp$ ensures that the selective parameters are non-negative, while $\alpha$ is a trainable scalar controlling the influence of $\Delta_t$. By employing this selective mechanism, FACTS is capable of compressing long sequences in its state-space memory while maintaining the key properties of latent permutation equivariance and row-wise permutation invariance. Hence, FACTS can efficiently capture meaningful factors, e.g., objects in video frames or independent sources in a signal, even as their relationships with input features change over time.

**Linearisation** Although the framework presented so far in equation 10 has a permutable state-space memory and equivariant through the memory-input routing, which we formally show in Section 3.2,

the routing between $Z_{t-1}$ and $X_t$ introduces dependencies of $\bar{B}(Z_{t-1}, X_t)$ and $U(Z_{t-1}, X_t)$ on $Z_{t-1}$. This results in a non-linear recurrence between $Z_{t-1}$ and $Z_t$ in equation 10, which limits parallelisation and hinders training efficiency. To overcome this, we substitute $Z_{t-1}$ with $Z_0$, i.e. the initial memory or state, within the information routing processes. This leads to the final formulation of FACTS:

$$Z_t = \textbf{FACTS}(Z_{t-1}, Z_0, X_t) \tag{16}$$

$$= \bar{A}(Z_0, X_t) \odot Z_{t-1} + \bar{B}(Z_0, X_t) \odot U(Z_0, X_t) \tag{17}$$

$$= \bar{A}(Z_0, X_t) \odot \textbf{FACTS}(Z_{t-2}, Z_0, X_{t-1}) + \bar{B}(Z_0, X_t) \odot U(Z_0, X_t) \tag{18}$$

$$= \sum_{s=0}^{t} \bar{A}^{\times}(Z_0, X_{t:s}) \odot \bar{B}(Z_0, X_s) \odot U(Z_0, X_s), \tag{19}$$

$$= \textbf{FACTS}(Z_0, X_{1:t}) \tag{20}$$

where $\bar{A}^{\times}(Z_0, X_{t:s}) = \bar{A}_{t+1} \odot \bar{A}_t \odot \bar{A}_{t-1} ... \odot \bar{A}_{s+1}$ and $\bar{A}_{t+1}$ is filled with "1"; the initial state $Z_0$ can be given a priori, learnable, or sampled from a prior distribution. This formulation in equation 17 linearise the recurrence in equation 10 by breaking the non-linear dependency between $Z_t$ and $Z_{t-1}$. That is, as shown in equation 20, the inputs interact only with the initial memory, enabling fast computation of $Z_t$ using equation 19, i.e., without recurrence. This significantly improves computational efficiency. Note that while $Z_0$ is denoted as the "initial state", it needs not represent the sequence's true start; long sequences can be segmented (chunked) for parallel computation within segments while maintaining sequential dependencies across the original sequence.

## 3.2 THEORETICAL ANALYSIS OF FACTS

Here, we formally proof the permutation equivariance and invariance properties of FACTS. We first formally define the two fundamental properties, namely *left permutation equivariant* (L.P.E.) and *right permutation invariant* (R.P.I.) in Definitions 1 and 2, respectively.

**Definition 1.** Let $f : \mathbb{R}^{n_1 \times n_2} \times \mathbb{R}^{t \times n_3 \times n_4} \to \mathbb{R}^{n_1 \times n_5}$ be a bivariate function with $n_1, n_2, n_3, n_4, n_5, t \in \mathbb{N}$. $f$ is permutation equivariant (L.P.E.) if for all $\sigma \in S_{n_1}$, $M_1 \in \mathbb{R}^{n_1 \times n_2}$, and $M_2 \in \mathbb{R}^{t \times n_3 \times n_4}$,

$$f(\sigma M_1, M_2) = \sigma f(M_1, M_2),$$

where $S_k$ denotes the set of permutation matrices of size $\mathbb{R}^{k \times k}$.

**Definition 2.** Let $\mathbb{R}^{n_1 \times n_2} \times \mathbb{R}^{t \times n_3 \times n_4} \to \mathbb{R}^{n_1 \times n_5}$ be a bivariate function with $n_1, n_2, n_3, n_4, n_5, t \in \mathbb{N}$. $f$ is right permutation invariant (R.P.I.) if for all $\sigma_1, \sigma_2, \ldots, \sigma_t \in S_{n_3}$, $M_1 \in \mathbb{R}^{n_1 \times n_2}$, and $M_2^1, M_2^2, \ldots, M_2^t \in \mathbb{R}^{n_3 \times n_4}$,

$$f(M_1, [\sigma_1 M_2^1, \sigma_2 M_2^2, \ldots, \sigma_t M_2^t]) = f(M_1, [M_2^1, M_2^2, \ldots, M_2^t]).$$

These L.P.E. and R.P.I. properties, which formally describe the two fundamental aspects of FACTS: *permutable memory* and *permutation-invariant recurrence (w.r.t. the features)* — with memory $Z_{t-1}$ and input $X_t$ serving as the left and right arguments of FACTS. They are thus essential not only for constructing the routing mechanism but also for the overall design of FACTS.

Using Definitions 1 and 2, that by taking memory $Z_t$ and features $X_t$ as the left and right arguments in FACTS (equation 20), we can show the following result:

**Theorem 1.** **FACTS** *as defined in equation 20 is L.P.E. and R.P.I.*

The proof of Theorem 1 is available in Appendix B. Theorem 1 proves our main claim that FACTS: i) is invariant to input features permutation. ii) learns permutable state-space memory. Furthermore, it is possible to extend our results in Theorem 1 to the more general case, where $\bar{A}, \bar{B}, U$ are L.P.E. and R.P.I. functions of $Z_{t-1}$ and $X_t$. The main result is presented in Theorem 2.

**Theorem 2.** *if $\bar{A}, \bar{B}, U$ are L.P.E. and R.P.I. functions of $Z_{t-1}$ and $X_t$, any dynamics governed by equation 10 is L.P.E. and R.P.I.*

The proof of Theorem 2 is available in Appendix B. Theorem 2 highlights the main condition on the variables $\bar{A}, \bar{B}, U$ to ensure that the model is invariant to input features and has an equivariant memory. This can spark future research to develop SSM models based on equation 10 that are efficient history compressors and are suitable to dynamic world modelling scenarios.

## 4 EXPERIMENTS

We design experiments to evaluate the effectiveness of FACTS in world modelling. We frame world modelling as a prediction task, where the model must predict future events in complex environments based on observed history, and we assess its performance by prediction accuracy. Experiments are conducted on three environments: the multivariate time series (MTS) benchmark for forecasting, synthetic multi-object videos (Yi et al., 2020; Greff et al., 2022; Lin et al., 2020), and dynamic-graph node prediction (Li et al., 2018). Additionally, Appendix D presents an extensive ablation study that highlights the robustness of FACTS.

### 4.1 LONG TERM MULTIVARIATE TIME-SERIES FORECASTING

In many real-world applications, such as climate prediction, traffic flow management, or autonomous systems, predicting future states over long horizons is crucial for effective decision-making. World modelling, in these domains, often involves high-dimensional multivariate inputs—such as interacting agents, variables, or environmental factors—requiring the system to account for their complex dependencies and interactions across time. Long-term forecasting in this context is challenging as it demands accurate representation of temporal dynamics over extended periods. Additionally, input features may lack a predefined order, or this order could change dynamically. For example, a system may receive data from sensors (e.g., temperature and pressure) without knowing which is which during testing. The world model must provide reliable long-term predictions even if the input order changes unexpectedly and generalise to unseen configurations, without requiring exposure to every possible permutation of input features during training.

We use the open-source Time Series Library (TSLib)[1] to evaluate long-term multivariate time-series forecasting (MSTF) tasks across 9 real-world datasets spanning multiple domains (e.g., energy, weather, and finance). FACTS is compared against 8 state-of-the-art baseline models (Wang et al., 2024c; Liu et al., 2024; Wu et al., 2023a; Nie et al., 2023; Zeng et al., 2023; Zhang & Yan, 2023; Zhou et al., 2022; Wu et al., 2021), following TSLib's standardised settings: the input sequence length is fixed at 96, with prediction lengths of $\{96, 192, 336, 720\}$. Performance is evaluated using mean-squared error (MSE) and mean-absolute error (MAE). C.f. Appendix C.2 for more details.

### 4.1.1 FORECASTING WITH PREDEFINED ORDER (SCENARIO I)

We use the exact same setup to Wang et al. (2024c); Liu et al. (2024); Wu et al. (2023a), with the exception of the pre- and post-processing modules (referred to as the "embedders" and "projectors" in TSLib). In our implementation, we replace these with set functions to accommodate the output structure of FACTS (c.f. Appendix C for more details). Note that in the standard setup of TSLib, the arrangement of the input features in the test is not changed and is the identical to the arrangement to the one seen during the training. The average results over the different prediction windows of our proposed approach along with all competing methods are presented in Table 1 and the full results are available in Table 9 in Appendix E.

***Results*** Table 1 highlights the strong performance of FACTS, which achieves competitive results in both metrics, compared to the competing state-of-the-art specialised MSTF models. As shown, FACTS achieves the best MAE and MSE on 6 out of 9 datasets and, in terms of MAE, is always in the top 2 in 8 out of 9 them. Even where it is not the top performer, FACTS remains highly competitive (2nd or 3rd place) as seen in Traffic and Solar-Energy underscoring its robustness and ability to adapt to different scenarios. FACTS' ability to efficiently capture long-term dependencies can be attributed to its structured state-space memory, which promotes the learning of modular patterns whose interactions can explain the spatial-temporal correlations of the multivariate observations.

**Parallel vs Recurrent FACTS** The parallelisation design outlined in our equation 17-20 not only enhances computational efficiency but also provides flexibility for recurrent applications of the FACTS core operations. As mentioned, long sequences can be segmented for parallel computation within segments while preserving sequential dependencies across the entire sequence. To clarify further, we analysed FACTS' long-term forecasting performance by comparing its fully sequential mode to its fully parallel mode, adjusted by segment window size, on the MTS Electricity dataset (c.f. Figure 3). Each window size represents a different update frequency for $Z_0$: for example, a window

---

[1]Time Series Library benchmark: `https://github.com/thuml/Time-Series-Library.git`

| | | ETTm1 | ETTm2 | ETTh1 | ETTh2 | Electricity | Exchange | Traffic | Weather | Solar-Energy |
|---|---|---|---|---|---|---|---|---|---|---|
| **Autoformer** (2021) | MSE | 0.588 | 0.327 | 0.496 | 0.450 | 0.227 | 0.613 | 0.628 | 0.338 | 0.885 |
| | MAE | 0.617 | 0.371 | 0.487 | 0.459 | 0.338 | 0.539 | 0.379 | 0.382 | 0.711 |
| **FEDformer** (2022) | MSE | 0.448 | 0.305 | **0.440** | 0.437 | 0.214 | 0.519 | 0.610 | 0.309 | 0.291 |
| | MAE | 0.452 | 0.349 | 0.460 | 0.449 | 0.327 | 0.429 | 0.376 | 0.360 | 0.381 |
| **TimesNet** (2023a) | MSE | 0.400 | 0.291 | 0.458 | 0.414 | 0.192 | 0.416 | 0.620 | 0.259 | 0.301 |
| | MAE | 0.406 | 0.333 | 0.450 | 0.427 | 0.295 | 0.443 | 0.336 | 0.287 | 0.319 |
| **PatchTST** (2023) | MSE | **0.387** | **0.281** | 0.469 | 0.387 | 0.205 | 0.367 | 0.481 | 0.259 | 0.270 |
| | MAE | 0.400 | 0.326 | 0.454 | 0.407 | 0.290 | 0.404 | 0.304 | 0.281 | 0.307 |
| **DLinear** (2023) | MSE | 0.403 | 0.350 | 0.456 | 0.559 | 0.212 | 0.354 | 0.625 | 0.265 | 0.330 |
| | MAE | 0.407 | 0.401 | 0.452 | 0.515 | 0.300 | 0.414 | 0.383 | 0.317 | 0.401 |
| **Crossformer** (2023) | MSE | 0.513 | 0.757 | 0.529 | 0.942 | 0.244 | 0.940 | 0.550 | 0.259 | 0.641 |
| | MAE | 0.496 | 0.610 | 0.522 | 0.684 | 0.334 | 0.707 | 0.304 | 0.315 | 0.639 |
| **iTransformer** (2024) | MSE | 0.407 | 0.288 | 0.454 | 0.383 | 0.178 | 0.360 | 0.428 | 0.258 | **0.233** |
| | MAE | 0.410 | 0.332 | 0.447 | 0.407 | 0.270 | 0.403 | 0.282 | 0.278 | **0.262** |
| **S-Mamba** (2024c) | MSE | 0.398 | 0.288 | 0.455 | 0.381 | 0.170 | 0.367 | **0.414** | **0.251** | 0.240 |
| | MAE | 0.405 | 0.332 | 0.450 | 0.405 | 0.265 | 0.408 | **0.276** | **0.276** | 0.273 |
| **FACTS (Ours)** | MSE | 0.392 | **0.281** | **0.440** | **0.373** | **0.166** | **0.342** | 0.472 | **0.251** | 0.253 |
| | MAE | **0.397** | **0.325** | **0.428** | **0.399** | **0.263** | **0.392** | 0.303 | 0.278 | 0.272 |

Table 1: Average MSE and MAE of the different approaches on the multivariate time series forecasting tasks. For each metric and each dataset, the top performance and the second best are highlighted in **red** and blue, respectively.

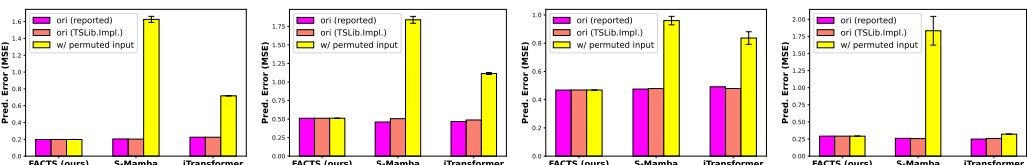

Figure 2: Model robustness to input permutations on 4 MTSF datasets (left to right: Electricity, Traffic, ETTm1, SolarEnergy). Magenta bars represent original performance, salmon bars show performance using our/TSLib implementation, and yellow bars represent results under input permutation. Results are averaged over five random seeds, with error bars showing ±2× standard deviation.

size of 1 corresponds to fully recurrent FACTS, while a window size of 96 (the sequence length) corresponds to fully parallel FACTS. As shown in Figure 3, the models consistently maintained strong performance across various segment window sizes.

### 4.1.2 FORECASTING WITH UNKNOWN ORDER (SCENARIO II)

To evaluate robustness under dynamic scenarios, we randomly permute the input features during the testing to simulate environments where the arrangement of agents or entities (e.g., robots, sensors) changes unpredictably. This mirrors real-world scenarios where input configurations vary, challenging world models to adapt to unseen input orderings. We focus on top pretrained models from the first scenario (Table 1), i.e., FACTS, iTransformer, and S-Mamba. For the datasets, we use the challenging ones from the first scenario and permute the feature embeddings five times during testing, reporting the average performance and two standard deviations.

***Results*** The main results are presented in Figure 2. While other models, iTransformer and S-Mamba, experience significant degradation in performance when the input feature orders are shuffled, FACTS consistently maintains its prediction performance across the different tasks. These findings corroborate the theoretical results of Section 3.2. For example, on the Traffic dataset, the MSE error of S-mamba, which is the model with the top performance in the standard setting (Table 1), increases more than threefold, and the error for iTransformer doubles. In contrast, FACTS, leveraging its selective memory routing which consistently assigns input features to the latent factors, preserves low error rates despite the permutation. This highlights FACTS' ability to handle dynamic and unordered environments. This adaptability further emphasizes the generalisation strength of FACTS, particularly in world modelling scenarios where input orders may be inconsistent or unknown.

### 4.2 OBJECT-CENTRIC WORLD MODELLING

Visual object-centric representation learning (OCRL) (Burgess et al., 2019; Greff et al., 2019; Nanbo et al., 2020) tackles the challenge of binding visual information to consistent factors, even as object

| Method | LPIPS ↓ | |
| --- | --- | --- |
| | CLEVRER | OBJ3D |
| PredRNN (2017) | 0.17 | 0.12 |
| VQFormer (2019) | 0.18 | 0.11 |
| G-SWM (2020) | 0.16 | 0.10 |
| SlotFormer (2023b) | 0.11 | 0.08 |
| SAVi-dyn (2023) | 0.19 | 0.12 |
| **FACTS (Ours)** | **0.09** | **0.07** |

Table 2: Quantitative results for slot dynamics prediction: visual quality of future frame prediction, measured by LPIPS (lower is better).

| | FG-ARI↑ | mIoU↑ |
| --- | --- | --- |
| SAVi (2023) | 0.64 | 0.43 |
| OC-SlotSSM (2024) | 0.68 | 0.55 |
| **FACTS (Ours)** | **0.75** | **0.60** |

Table 3: Quantitative results for unsupervised object discovery: segmentation quality on MOVI-A under the video reconstruction setting, evaluated by FG-ARI and mIoU (higher indicate better).

features dynamically permute with movement across pixels in videos (Kipf et al., 2023). This aligns with FACTS' objective of identifying regularities in dynamic environments for history compression and future-event prediction, making OCRL an ideal evaluation benchmark. To evaluate how FACTS 1) leverages object information for future predictions and 2) aligns its discovered factors with objects, we conduct two OCRL experiments, *slot dynamics prediction* and *unsupervised object discovery*, set on widely-used OCRL datasets (Yi et al., 2020; Lin et al., 2020; Greff et al., 2022).

**Slot Dynamics Prediction** The task involves having a world model capture object-centric dynamics in latent space: given the latent object representations of observed events ("burn-in"), the model predicts the future latent codes of the objects ("roll-out"). We conducted this experiment following the setup of our major baseline, SlotFormer (Wu et al., 2023b). We evaluate the performance of the model by assessing 1) the visual quality of the predicted future frames and 2) the precision of future segmentation map rendered from the predicted latents. We quantify visual quality using the LPIPS metric, which provides stronger alignment with human perception than other commonly used metics such as PSNR and SSIM (Wu et al., 2023b; Sara et al., 2019), and segmentation accuracy using the commonly used Mean Intersection over Union (mIoU) and Adjusted Rand Index (ARI), with and/or without the foreground focus.

*Results* The results presented in Table 2 and Table 8 highlight the strengths of the FACTS model in terms of both visual quality and segmentation accuracy for object dynamics prediction on the CLEVRER dataset. FACTS achieves the lowest LPIPS score of 0.09, indicating superior visual quality in the predicted frames. Additionally, it achieves competitive segmentation accuracy, achieving a leading FG-mIoU of $48.11\%$. This highlights its effectiveness in predicting object positions and interactions in future frames. We attribute these results to FACTS' selective history compression mechanism. In contrast to SlotFormer, which predicts the next state by attending to all past inputs—resulting in inefficiency and noisy predictions—FACTS effectively compresses and retains only the most relevant information in memory, thereby filtering out noise and enabling more accurate dynamics modelling and future predictions.

**Unsupervised Object Discovery** In contrast to the slot dynamics task, where object slots or factors are given as input, this experiment requires FACTS to automatically discover relevant factors for future predictions in multi-object videos. This process enables us to understand the regularities that FACTS conceptualises as significant for forecasting future events. We utilised a CNN encoder to convert the input image into a feature set, from which FACTS learns the object factors. These factors are employed to predict future object slots and are subsequently decoded back into video frames using a spatial-broadcast decoder. Note that, in this experiment, we adopted the fully-unsupervised setup of SAVi and jointly train all the modules (including the CNN encoder and spatial-broadcast decoder) end-to-end from scratch by minimising the reconstruction MSE and future prediction MSE. In addition to future prediction, we also conducted object discovery under the video reconstruction setting using the MOVi-A dataset to enable direct comparison with the state-of-the-art video-based OCRL models for videos, i.e., SAVi (Kipf et al., 2023) and OC-SlotSSM (Jiang et al., 2024).

*Results* We visualise the discovered factors by independently rendering each factor's dynamics back into videos. In the "discovery for prediction" task, FACTS primarily identifies moving objects - considered "useful" for future predictions—while treating static objects as background. In contrast, for the "discovery for reconstruction" task, FACTS identifies also static objects as explanatory factors (see Figure 8)—like SAVi and OC-SlotSSM. We attribute this behaviour to the residual design

| Method | RMSE↓ | MAE↓ | MAPE↓ |
|---|---|---|---|
| HA (2018) | 12.16 | 5.92 | 15.17% |
| LSTNet (2018) | 9.22 | 5.11 | 12.56% |
| STGCN (2018) | 7.92 | 3.87 | 10.05% |
| DCRNN (2018) | 7.87 | 3.85 | 10.01% |
| GWN (2019) | 7.66 | 3.54 | 9.98% |
| ASTGCN (2019) | 7.99 | 3.94 | 10.12% |
| GMA (2020) | 8.32 | 4.06 | 10.91% |
| MTGNN (2020b) | 8.16 | 3.99 | 10.28% |
| AGCRN (2020) | 8.22 | 4.02 | 10.53% |
| DGCRN (2023) | 7.19 | 3.44 | 9.73% |
| STGM (2023) | 7.10 | 3.23 | 9.39% |
| MegaCRN (2023) | 7.23 | 3.38 | 9.72% |
| TESTAM (2024) | 7.09 | 3.36 | 9.67% |
| **FACTS (Ours)** | **6.97** | **3.11** | **9.08%** |

Table 4: RMSE, MAE, and MAPE of various approaches for long-term graph node forecasting (1-hour, 12-step ahead) on the METR-LA dataset.

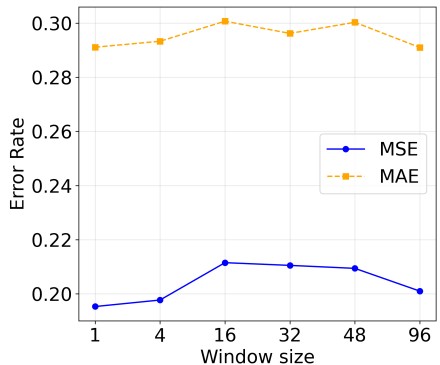

Figure 3: Parallel vs Recurrent FACTS on long-term MTS (Electricity): MSE and MAE for different window/chunk sizes. Input (observable) sequence length set to 96.

of the FACTS predictor, which is muted during reconstruction. Our quantitative results in Table 3 demonstrate that FACTS outperforms both OC-SlotSSM and SAVi in unsupervised object discovery (for reconstruction), highlighting its superior performance despite being a more general framework. Additional visual results of our object-centric world modelling are available in the Appendix E.

### 4.3 LONG TERM PREDICTION WITH GRAPH DATA

To demonstrate FACTS' flexibility as a powerful general framework able to handle different input types/modalities and efficiently solve diverse forecasting-based tasks, we also apply FACTS to dynamic-graph input data evaluated on long-term node prediction task (12-step MAE) with the commonly used METR-LA dataset (Li et al., 2018) and we compare against existing state-of-the-art approaches on this task. We refer to Appendix C.4 for more experimental details.

***Results*** As can be seen in Table 4. FACTS, leveraging its graph-structured memory, also outperforms all existing methods on this task, even those specialised for this task. For instance, TESTAM (Lee & Ko, 2024) yields an MAPE of $9.67\%$ whereas FACTS yields $9.08\%$. This further corroborates our main claim that FACTS is indeed a versatile world model framework with consistently strong performance in several diverse forecasting tasks.

## 5 DISCUSSIONS & CONCLUSION

In this work, we introduced FACTS, a novel recurrent framework designed for spatial-temporal world modelling. FACTS is constructed permutable state-space memory, which offers the flexibility needed to capture complex dependencies across time and space. By employing selective memory-input routing, FACTS is able to dynamically assign input features to distinct latent factors, enabling more efficient history compression and long-term prediction accuracy. Furthermore, we formally showed that FACTS: i) is invariant to input feature permutation. ii) learns permutable state-space memory, maintaining consistent factor representations regardless of changes in the input order. Furthermore, through comprehensive empirical evaluations, FACTS demonstrated superior performance on a variety of real-world datasets, consistently matching or outperforming specialised state-of-the-art models in diverse tasks. Notably, FACTS maintained its predictive powers even in challenging settings where the order of input features was shuffled, highlighting its robustness and adaptability. These results underscore the model's potential for a wide range of applications, particularly in world modelling scenarios where input configurations are variable or uncertain. For future work, we plan to extend FACTS to larger-scale experiments, exploring its scalability and potential in even more complex world modelling tasks.

ACKNOWLEDGEMENT

The research reported in this publication was supported by funding from King Abdullah University of Science and Technology (KAUST) - Center of Excellence for Generative AI, under award number 5940.

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

# A    RELATED WORKS

From a historical perspective, "world models" or using models to learn environmental dynamics and leveraging them in policy training has an extensive literature, with early foundations laid in the 1980s using feed-forward neural networks (Werbos, 1987; 1989; Munro, 1987; Robinson & Fallside, 1989; Nguyen & Widrow, 1990) and in the 1990s with RNNs (Schmidhuber, 1990b;a; 1991b;a). Notably, PILCO (Deisenroth & Rasmussen, 2011; McAllister & Rasmussen, 2016) has emerged as a key probabilistic model-based method, using Gaussian processes (GPs) (MacKay et al., 1998) to learn system dynamics from limited data and train controllers for tasks like pendulum swing-up and uni-cycle balancing. While GPs perform well with small, low-dimensional datasets, their computational complexity limits scalability in high-dimensional scenarios. To address this, later works (Gal et al., 2016; Depeweg et al., 2016) have adopted Bayesian neural networks (Kononenko, 1989), which have demonstrated success in control tasks with well-defined states (Hein et al., 2017). However, these methods remain limited when modelling high-dimensional environments, such as sequences of raw pixel frames. In the context of reinforcement learning, using recurrent models to learn system dynamics from compressed latent spaces has significantly improved data efficiency (Schmeckpeper et al., 2020; Finn et al., 2016). While the development of internal models for reasoning about future states using RNNs dates back to the early 1990s, subsequent works, such as "Learning to Think" (Schmidhuber, 2015) and "World Models" (Ha & Schmidhuber, 2018), have extended this by introducing RNN-based frameworks that model environments and reason about future outcomes. These RNN-based models have been applied to future frame generation (Chiappa et al., 2017; Oh et al., 2015; Denton et al., 2017) and reasoning about future outcomes (Silver et al., 2017; Watters et al., 2017). However, as RNNs suffer from the vanishing gradients problem (Hochreiter, 1991; Pascanu et al., 2013), recently there has been a growing interest in using Transformers (Chen et al., 2022; Robine et al., 2023; Micheli et al., 2022) and SSM-based appraoches (Gu & Dao, 2023; Hafner et al., 2023; Samsami et al., 2024) for world modelling.

As world modelling is fundamentally intertwined with sequence modelling (Schmidhuber, 1990b), it often carries temporal implications that align with the principle of history compression (Schmidhuber, 1992b; 2003). Temporal selectivity is essential in these models, with Recurrent Neural Networks (RNNs), particularly those with gating mechanisms like LSTMs (Hochreiter & Schmidhuber, 1997), GRUs (Cho, 2014), and xLSTMs (Beck et al., 2024), being well-suited for this task. However, learning from high-dimensional sequential data complicates the problem, posing a core challenge in spatial-temporal learning. This challenge is exacerbated by the quadratic computation scaling in transformers, despite their success. Approaches like dimensionality reduction (Hotelling, 1933; Tipping & Bishop, 1999; Kingma, 2013) and predictability minimisation (Schmidhuber, 1992c; Ghahramani, 1994) must adhere to the principle of history compression along the temporal axis, rather than compressing spatial information at each time step independently. From the perspective of information bottleneck principle (Tishby et al., 2000), the goal is to selectively extract the "bottleneck" from high-dimensional sequences that is most useful for world modelling tasks like predicting future events.

Recently, the emergence of Mamba (Gu & Dao, 2023) and other SSM-based frameworks (Gu et al., 2021; Dao & Gu, 2024; Wang et al., 2024b) has garnered widespread attention for their strong performance in efficient sequence modelling. Mamba structure is similar to LSTM (Hochreiter & Schmidhuber, 1997), in the sense that it utilises a forget gate, an input gate, and an output gate. The key difference is that these gates depend only on the previous input (not on the hidden state representing the history of inputs so far). While this hinders their representation power (e.g., cannot solve the parity problem (Hochreiter & Schmidhuber, 1996; Schmidhuber et al., 2007; Srivastava et al., 2015)), this formulation enables parallel computation of selective history compression via sub-linear sequential attention (Dao & Gu, 2024), constructing dependencies between distant data points within the sequence. This sparks their successful applications across various tasks including language modelling (Mehta et al., 2022; Grazzi et al., 2024; He et al., 2024), deep noise suppression (Du et al., 2024b), and clinical note understanding (Yang et al., 2024b). Additionally, many SSM-based vision models have been proposed for tasks such as classification (Du et al., 2024a; Shi et al., 2024; Baron et al., 2023; Huang et al., 2024; Smith et al., 2023; Nguyen et al., 2022), detection (Chen et al., 2024), segmentation (Yang et al., 2024a; Ma et al., 2024), generation (Yan et al., 2024; Fei et al., 2024), and video understanding (Islam & Bertasius, 2022; Wang et al., 2023). Despite their success, existing SSMs often lack efficient mechanisms for handling high-dimensional

spatial data, relying primarily on linear and rigid structural biases (Gu et al., 2021; Dao & Gu, 2024), particularly when dealing with permutable spatial structures.

Multivariate time-series forecasting (MTSF) and object-centric representation learning (OCRL) both involve working with noisy, high-dimensional sequential data, making them critical benchmarks for evaluating world models with significant real-world impact. We chose to assess our models on these tasks because they exemplify the primary challenge of spatial-temporal learning. Existing MTSF methods either struggle to effectively model long-term temporal dependencies (Salinas et al., 2020; Zhang & Yan, 2023; Nie et al., 2023) or fail to effectively leverage the cross-variate regularities in high-dimensional inputs (Wu et al., 2021; Zhou et al., 2022; Wu et al., 2023a; Zeng et al., 2023; Wang et al., 2024a), resulting in inaccuracies in future-state forecasting. Two notable models in the field, iTransformer (Liu et al., 2024) and S-Mamba (Wang et al., 2024c), also have limitations. iTransformer suffers from the quadratic scaling of transformers, making it difficult to capture long-term dependencies, while S-Mamba struggles with handling the spatial structures of the data. Object-centric representations, or "slots", are designed to capture "objects", i.e. solving the binding problem (Greff et al., 2020). Our goal is more general: we aim to capture modularities, or "factors", that remain invariant across sequences, framing the discovery of spatial regularities in history compression as another instance of the binding problem. Although a philosophical discussion on whether these "factors" should align with common-sense "objects" is beyond the scope of this paper, OCRL is closely related and serves as a good demonstration of our approach. OCRL originated from the *vision-as-inverse-Bayes* framework (Yuille & Kersten, 2006), initially applied to images (Burgess et al., 2019; Greff et al., 2019; Locatello et al., 2020), later extended to videos (Nanbo et al., 2020; Kipf et al., 2023), and developed into object-centric world models (Lin et al., 2020; Kipf et al., 2019; Wu et al., 2023b; Stanić et al., 2023). Recent OCRL works heavily rely on the Slot Attention (SA) mechanism (Locatello et al., 2020) for object discovery, which is closely related to our routing modules. We view the SA, which also satisfies the LPE and RPI properties, as a suitable but computationally expensive alternative to equation 12. In addition to MTSF and OCRL, because FACTS leverages its graph-structured memory for spatial-temporal prediction, it is also closely related to geometric modelling and learning (Bronstein et al., 2017; Wu et al., 2020a). In this paper, we demonstrate FACTS' potential for handling non-Euclidean data through long-term graph node prediction experiments.

It is worth mentioning Goyal et al. (2020; 2022), which propose latent state factorisation and equivariance in attention-augmented LSTM/GRU-based frameworks. However, unlike these works, FACTS employs structured state-space memory, enabling dynamic input-to-factor assignments with explicit permutation invariance and efficient training. Recently, SlotSSMs (Jiang et al., 2024) introduced factorisation into SSMs by stacking slot attentions on top of Mamba blocks, adding an extra layer to handle modular patterns. In contrast, FACTS offers a fundamentally different state-space formulation, seamlessly integrating factorisation, geometric modelling, and selectivity within a single block through dynamic memory-input routing. While SlotSSMs can be viewed as a special case of FACTS under specific conditions (akin to the fully parallel FACTS), FACTS' versatile architecture ensures robust performance across diverse tasks and evolving environments without task- or modality-specific modifications.

## B PROOFS

**Theorem** (Restatement of Theorem 1). **FACTS** *as defined in equation 20 is L.P.E. and R.P.I.*

*Proof.* Let $\sigma_Z \in S_k$, $\sigma_X^1, \sigma_X^2, \ldots, \sigma_X^t \in S_m$, $Z_0, Z_t \in \mathbb{R}^{k \times d}$, $X_1, X_2, \ldots, X_t \in \mathbb{R}^{m \times d}$. By equation 17, equation 18 equation 19, and equation 20, it is sufficient to show

$$\sigma_Z \mathbf{FACTS}(Z_k, Z_0, X_k) = \mathbf{FACTS}(\sigma_Z Z_k, \sigma_Z Z_0, \sigma_X^t X_k).$$

for all $k \in \mathbb{N}, k \in [1, t]$.

$$\sigma_Z \mathbf{FACTS}(Z_k, Z_0, X_k)$$
$$= \sigma_Z(\bar{A}(Z_0, X_k) \odot Z_{k-1} + \bar{B}(Z_0, X_k) \odot U(Z_0, X_k))$$
$$= \sigma_Z \bar{A}(Z_0, X_k) \odot \sigma_Z Z_{k-1} + \sigma_Z \bar{B}(Z_0, X_k) \odot \sigma_Z U(Z_0, X_k).$$

---

**Algorithm 1 FACTS** Module: a Pseudo Implementation

---

1: **Input:** $X_{1:t} \in \mathbb{R}^{t \times m \times d}$          $\triangleright$ $t$-sequential axis, $m$-spatial axis
2: **Output:** $Z_{1:t} \in \mathbb{R}^{t \times k \times d}$
3: *Initialise:* $Z_0 \in \mathbb{R}^{k \times d}$          $\triangleright$ E.g., init using learnables, or Gaussian samples
4: *Param:* $\boldsymbol{A} = \alpha \in \mathbb{R}$
5:
6: $X_{1:t} \leftarrow \text{Rearrange}(\text{Linear}(X_{1:t}), [t, m, p] \to [p, t, m])$      $\triangleright$ Input projection
7: $(X_{1:t}, \Delta_{1:t}, \boldsymbol{B}_{1:t}, \boldsymbol{C}_{1:t}) \leftarrow \text{Conv2D}(X_{1:t}; \text{ksz}=(\text{dconv}, 1)).\text{split}(\text{axis}=1)$    $\triangleright$ Conv along $t$-axis
8: $(\boldsymbol{U}_{1:t}, \Delta_{1:t}, \boldsymbol{B}_{1:t}, \boldsymbol{C}_{1:t}) \leftarrow \text{Rout}(Z_0, (X_{1:t}, \Delta_{1:t}, \boldsymbol{B}_{1:t}, \boldsymbol{C}_{1:t}))$    $\triangleright$ Routing for the elements[2]
9: $(\bar{\boldsymbol{A}}_{1:t}, \bar{\boldsymbol{B}}_{1:t}) \leftarrow \text{Discretisation}(\text{Softplus}(\Delta_{1:t}), \boldsymbol{A}_{1:t}, \boldsymbol{B}_{1:t})$
10: $Z_{1:t} \leftarrow \text{StateSpacePropagation}(Z_0, \boldsymbol{U}_{1:t}, \bar{\boldsymbol{A}}_{1:t}, \bar{\boldsymbol{B}}_{1:t})$     $\triangleright$ $Z_{1:t} \in \mathbb{R}^{t \times k \times d}$, c.f. equation 17-19
11: $\hat{Z}_{1:t} \leftarrow \boldsymbol{C}_{1:t} \odot Z_{1:t}$         $\triangleright$ $\hat{Z}_{1:t} \in \mathbb{R}^{t \times k \times d}$ for selective output, e.g. $\text{Dec}(\hat{Z}_{1:t})$
12:
13: **Return:** $\hat{Z}_{1:t}, Z_{1:t}$         $\triangleright$ $\hat{Z}_{1:t}$: the output; $Z_{1:t}$: the state representation

---

Since $\bar{A}, \bar{B}, U$ are L.P.E. and R.P.I.,

$$\sigma_Z \, \textbf{FACTS}(Z_k, Z_0, X_k)$$
$$= \sigma_Z \bar{A}(Z_0, X_k) \odot \sigma_Z Z_{k-1} + \sigma_Z \bar{B}(Z_0, X_k) \odot \sigma_Z U(Z_0, X_k)$$
$$= \bar{A}(\sigma_Z Z_0, \sigma_X^k X_k) \odot \sigma_Z Z_{k-1} + \bar{B}(\sigma_Z Z_0, \sigma_X^k X_k) \odot U(\sigma_Z Z_0, \sigma_X^k X_k)$$
$$= \textbf{FACTS}(\sigma_Z Z_k, \sigma_Z Z_0, \sigma_X^t X_k).$$

$\square$

**Theorem** (Restatement of Theorem 2). *if* $\bar{A}, \bar{B}, U$ *are L.P.E. and R.P.I. functions of* $Z_{t-1}$ *and* $X_t$, *any dynamics governed by equation 10 is L.P.E. and R.P.I.*

*Proof.* Let $Z_{t-1} \in \mathbb{R}^{k \times d}, X_t \in \mathbb{R}^{m \times d}, \sigma_Z \in S_k$, and $\sigma_X \in S_m$ be matrices. Assume $\bar{A}, \bar{B}, U$ are L.P.E. and R.P.I. functions of $Z_{t-1}$ and $X_t$. By expanding equation 10,

$$\bar{A}_t(\sigma_Z Z_{t-1}, \sigma_X X_t) \odot \sigma_Z Z_{t-1} + \bar{B}_t(\sigma_Z Z_{t-1}, \sigma_X X_t) \odot U_t(\sigma_Z Z_{t-1}, \sigma_X X_t) \tag{21}$$
$$= \sigma_Z \bar{A}_t(Z_{t-1}, X_t) \odot \sigma_Z Z_{t-1} + \sigma_Z \bar{B}_t(Z_{t-1}, X_t) \odot \sigma_Z U_t(Z_{t-1}, X_t) \tag{22}$$
$$= \sigma_Z (\bar{A}_t(Z_{t-1}, X_t) \odot Z_{t-1} + \bar{B}_t(Z_{t-1}, X_t) \odot Z_{t-1} \odot U_t(Z_{t-1}, X_t)) \tag{23}$$
$$= \sigma_Z Z_t \tag{24}$$

$\square$

## C    IMPLEMENTATION DETAILS

### C.1    FACTS MODULE

We provide the pseudo implementation of FACTS in Algorithm 1, and refer to our ⊕ Github page for more details. All results reported for FACTS in this paper were generated using a single NVIDIA A100 GPU (80 GB).

### C.2    LONG-TERM MULTIVARIATE TIME-SERIES FORECASTING

**Datasets** We use the open-source Time Series Library (TSLib), a widely-used benchmark for training and evaluating time-series models. TSLib provides standardized settings and a leaderboard of top-performing models, ensuring fair and consistent comparisons. Our focus is on long-term multivariate time-series forecasting (MSTF) tasks. We use a collection of 9 diverse real-world datasets, as presented in Table 5. These datasets span various domains, including energy production/consumption, finance exchange rate, traffic monitoring, and weather forecasting, offering a diverse range of variates and time granularities. The temporal resolutions of the datasets range from

---

[2] We omit the reshaping operation of the Conv2D output tensors ($[*, t, m] \to [t, m, *]$) for simplicity.

minutes to days, making them ideal for evaluating the performance of forecasting models over both short-term and long-term horizons.

Table 5: The datasets for MTSF evaluation.

| Datasets | Variates | Timesteps | Granularity |
|---|---|---|---|
| ETTm1 & ETTm2 | 7 | 17,420 | 15min |
| ETTh1 & ETTh2 | 7 | 69,680 | 1hour |
| Electricity | 321 | 26,304 | 1hour |
| Traffic | 862 | 17,544 | 1hour |
| Exchange | 8 | 7,588 | 1day |
| Weather | 21 | 52,696 | 10min |
| Solar-Energy | 137 | 52,560 | 10min |

**Baselines** We compare FACTS against 8 baseline models, including state-of-the-art MSTF approaches that top the TSLib leaderboard (Wang et al., 2024c; Liu et al., 2024; Wu et al., 2023a; Nie et al., 2023; Zeng et al., 2023; Zhang & Yan, 2023; Zhou et al., 2022; Wu et al., 2021). Below is a brief summary of 5 most notable baseline approaches:

- *S-Mamba* (Wang et al., 2024c): This baseline adapts Mamba models for MTS data by utilising a bidirectional scan on variates, achieving superior results compared to the previous leading method, iTransformer.

- *iTransformer* (Liu et al., 2024): An inverted transformer architecture that captures univariate history and cross-variate dependencies within a look-back window, though limited by the quadratic scaling of transformers. iTransformer has been leading the long-term forecasting task

- *TimesNet* (Wu et al., 2023a): Specialises in modelling multi-periodicity and interactions among periodic signals in MTS data.

- *CrossFormer* (Zhang & Yan, 2023): The emphasis is on modelling cross-dimension (spatial) interactions within MTS data.

- *PatchTST* (Nie et al., 2023): Uses patching techniques to segment sub-time sequences and model channel-wise transitions, improving temporal modelling.

**FACTS for MTSF** Due to the noisy nature of raw input data, a single time step (represented as a multivariate vector) often carries limited meaningful information. A common approach to handle this is to introduce feature encoders to pre-process the data—as those temporal and positional embedders used in the baselines. We employ a set embedder to map the input multivariate sequences of size $t \times m$ into $t \times m \times d$, augmenting the tensor with an additional dimension that allows each time step to be represented as a set of $m$ features, each of $d$-dimensional size. This resulting tensor, $t \times m \times d$, serves as the direct input to the FACTS model. For prediction, we adhere to the standard practice in the Time Series Library (TSLib) benchmark, which treats time-series models as encoders designed for single-step predictions, rather than auto-regressive forecasting. We show the MTSF process of FACTS in Algorithm 2. Note that our decoder, namely "factor-graph decoder",

---

**Algorithm 2** FACTS for Multivariate Time Series Forecasting

1: **Input:** $\boldsymbol{x}_{1:t} \in \mathbb{R}^{t \times m}$
2: **Output:** $\boldsymbol{x}_{t+1:t+f} \in \mathbb{R}^{f \times m}$
3:
4: $X_{1:t} \leftarrow \text{SetEmbedder}(\boldsymbol{x}_{1:t})$ $\qquad\qquad\qquad$ ▷ $X_{1:t} \in \mathbb{R}^{t \times m \times d}$, pre-normalisation, embedding
5: $Z_{1:t} \leftarrow \textbf{FACTS}(X_{1:t})$ $\qquad$ ▷ $Z_{1:t} \in \mathbb{R}^{t \times k \times d}$, **FACTS**() an encoder encapsulating Algo. 1
6: $Z_{t+1:t+f} \leftarrow \text{Predictor}(Z_{1:t})$ $\qquad$ ▷ $Z_{t+1:t+f} \in \mathbb{R}^{f \times k \times d}$, where $f$ is the prediction length
7: $\boldsymbol{x}_{t+1:t+f} \leftarrow \text{FactorGraphDecoder}(Z_{t+1:t+f})$ $\qquad\qquad\qquad$ ▷ $\boldsymbol{x}_{t+1:t+f} \in \mathbb{R}^{f \times m}$
8: $\boldsymbol{x}_{t+1:t+f} \leftarrow \boldsymbol{x}_{t+1:t+f} + \text{ResidualPred}(\boldsymbol{x}_{1:t})$ $\qquad$ ▷ Predictive residual $\in \mathbb{R}^{f \times m}$ feed-through
9: $\boldsymbol{x}_{t+1:t+f} \leftarrow \text{PostProcessing}(\boldsymbol{x}_{t+1:t+f})$ $\qquad\qquad$ ▷ Invert pre-normalisation
10:
11: **Return:** $\boldsymbol{x}_{t+1:t+f}$

---

is crucially designed to be invariant to the permutation of factors and processes the latent factors in parallel. Each factor independently makes predictions without relying on others. Specifically, the decoder aggregates the individual predictions of each factor by applying a softmax-weighted sum, ensuring that the final prediction effectively combines the contributions of all factors while maintaining permutation invariance. We details the aforementioned modules, i.e. the three embedders and the factor-graph decoder in the following list:

- **Discrete-Fourier Transform (DFT) decomposer**: this embedder applies the Fast Fourier Transform to each variate and decompose it into multiple spectral components. The top-$k$ low-frequency signals are treated as the "trend component", while the high frequency signals represent the "seasonal component". By concatenating the "trend" and "seasonal" components, we embed each univariate signal at each time point into a 2D (or $k+1$ dimensional, if the top-$k$ frequencies are not combined) vector representation, resulting in a set output that is compatible with FACTS.

- **Conv2d embedder**: this embedder applies a convolution along the temporal axis of the input data to aggregate information from nearby time points (i.e., "local context", with the look-back window controlling the kernel size. We implement this using PyTorch's standard Conv2d module, setting the kernel size to $[\text{lookback}, 1]$ and padding with zeros at the initial time steps. Although such embedders learn the input-feature mapping directly from data, making them flexible in capturing different relationships in different datasets, the choice of look-back window size is often intuitive.

- **Multi-scale Conv2d (MS-Conv2d) Embedder**: this embedder retains the learning flexibility of a Conv2d embedder while extending the single look-back window to multiple scales. By combining different scales, it captures features at varying granularities, making it the most robust among the three embedders and consistently delivering strong results (as shown in Table 6).

- **Factor Graph Decoder (FGD)**: this decoder takes in the a set of predicted latents $Z_f = \{z_f^i\}_{i=1:k}$ (c.f. Algorithm 2 for its definition) and first project each $z_f^i \in \mathbb{R}^d$ (can run parallel) to $\tilde{\alpha}_f^i \in \mathbb{R}^m$ (the logits) and $\tilde{x}_f^i \in \mathbb{R}^m$ (the prediction of the i-th factor). Then the $k$ logits, which correspond to the $k$ factor predictions, will be processed to $k$ categorical probabilities by a soft-max function: $\alpha_f^i = softmax(\tilde{\alpha}_f^i, \{\tilde{\alpha}_f^j\}_{j \neq i})$. The output prediction is the weighted sum of the factor predictions w.r.t. their corresponding probabilities as: $x_f = \sum_i^k \alpha_f^i \odot \tilde{x}_f^i$, similar to the processes of spatial mixing and alpha blending in the vision and graphics communities (Porter & Duff, 1984; Williams & Titsias, 2004; Greff et al., 2017). In vision tasks, the Spatial Broadcast Decoder (SBD, Watters et al. (2019)), which shares similar properties with FGD but is more computationally expensive, is a more commonly used option.

### C.3 OBJECT-CENTRIC WORLD MODELLING

**Benchmark** We conducted both the *slot dynamics prediction* on the CLEVRER (Yi et al., 2020) and OBJ3D (Lin et al., 2020) datasets, and the *unsupervised object discovery* experiments on both the CLEVRER (Yi et al., 2020) and MOVi-A (Greff et al., 2022) datasets. These datasets all consist of synthetic vision data and are used as standard benchmark for OCRL research.

**Slot-dynamics prediction** For the slot dynamics prediction task, we follow the setup in (Wu et al., 2023b), filtering out video clips with new objects entering the scene during the rollout period to ensure a consistent evaluation setting. The input to FACTS is the latent object representations extracted using a pre-trained object-slot encoder (SAVi, Kipf et al. (2023)) from video frames. That is, we consider the output of the pre-trained SAVi our data in such task – same as (Wu et al., 2023b). We extract the input latent object representations using a pre-trained object-slot encoder (SAVi Kipf et al. (2023)) from video frames, and train an auto-regressive roll-outer with a single FACTS layer to predict the latent representations for the next 10 frames, based on the latent codes from 6 observed frames. During testing, to ensure a fair comparison with SlotFormer, we burn-in the first 6 frames and roll out (predict) 48 frames. The predicted object representations are visualised using a pre-trained, frozen SAVi decoder (a spatial broadcast decoder, Wu et al. (2023b)) to render video frames.

**Unsupervised object discovery** For unsupervised object discovery, we follow the fully-unconditional setting of SAVi, using unbiased slot initialisation and relying solely on RGB video

| Pred.Len | MS-Conv2d Emb. | Conv2d Emb. | DFT Emb. | Avg±Std.Err. |
|---|---|---|---|---|
| 96 | 0.143 | 0.144 | 0.147 | 0.145±0.002 |
| 192 | 0.158 | 0.159 | 0.161 | 0.159±0.001 |
| 336 | 0.171 | 0.170 | 0.171 | 0.171±0.000 |
| 720 | 0.198 | 0.228 | 0.236 | 0.219±0.015 |

Table 6: Ablation: MSTF Performance of FACTS vs. different embedders (MSE↓).

| $k \setminus d$ | 1 | 128 | 512 | Avg.±Std.Err. |
|---|---|---|---|---|
| 1 | 0.349 | 0.330 | 0.328 | 0.336±0.007 |
| 3 | 0.349 | 0.337 | 0.326 | 0.337±0.007 |
| 5 | 0.349 | 0.331 | 0.349 | 0.343±0.006 |
| 7 | 0.349 | 0.331 | 0.330 | 0.337±0.006 |
| 9 | 0.349 | 0.332 | 0.352 | 0.344±0.006 |
| Avg.±Std.Err. | 0.349±0.000 | 0.332±0.001 | 0.336±0.006 | —— |

Table 7: Ablation: MSTF Performance of FACTS vs. (#factors $k$, #dimensions $d$) (MSE↓).

frames as input, with no additional information. The primary modification is replacing SAVi's recurrent slot attention modules with FACTS. Importantly, all of the used modules (CNN vision encoders, FACTS, and decoders) *end-to-end* in a single run without any supervision. We evaluated object discovery under two settings, i.e. video reconstruction and future-frame prediction. For both setting, the input number of frames are set to 6, while for prediction FACTS is asked to predict, in addition to the observed 6 frames, future 10 frames. The prediction setting is conducted with a loss function and a *predictive residual design* to emphasise future frame prediction accuracy. The predicted noise will be added to the decoded prediction, i.e., in the pixel space of the future frames. A similar design is also used in our MSTF experiments, c.f. the **ResidualPred** in Algorithm 2 (line 8). Such predictive residual designs allow FACTS to avoid handling pixel-level noise, enabling it to focus on environment dynamics more effectively. In our OCRL experiments, we define such a design as a mapping between input $t$-frame video to the pixel-wise noise of $f$ future video frames, denoted as **PixelResPredictor** : $t \times c \times h \times w \to f \times 1 \times h \times w$. As mentioned, in the future prediction experiments, we found this quite effective in capturing video dynamics as it sets free the slots (latent states) from capturing objects that are "irrelevant" for modelling their dynamics—i.e., the static objects. The static objects will be merged into the "background". In object reconstruction experiments, object reconstruction experiments, this residual predictor is turned off to ensure a fair comparison.

### C.4 Dynamic Graph Node Prediction

**Benchmark** We conducted experiments on the METR-LA traffic dataset, which is a benchmark data set for traffic prediction, capturing 207 county highway traffic sensor speed observations in Los Angeles metropolitan area. It contains traffic data collected from sensors placed on road segments, represented as a dynamic graph with nodes corresponding to sensors and edges capturing traffic correlations.

**Setup** We follow the experimental setup of STGM and incorporate FACTS with a masking mechanism. Specifically, masking is applied to the routing processes described in equation 12, which underpins the construction of selective state-space model (SSM) parameters, $\bar{A}$ and $\bar{B}$. FACTS captures this mask information and integrates it into state-space dynamics modelling, ensuring that only relevant historical data informs the predictions.

## D Ablation Study

FACTS requires the use of a set embedder and the predefined selection of the number of factors in the state-space memory prior to training. Our ablation study aims to investigate the impact of different set embedders and the choice of the number of predefined factors on model performance.

**Impact of different set encoders** We conduct our experiments on the Electricity dataset, testing four prediction lengths: 96, 192, 336, and 720. Three different set encoders are evaluated, each

employing different priors: a Discrete-Fourier Transform (DFT) decomposer, a trainable Conv2d embedder, and a multi-scale Conv2d embedder (inspired by Wu et al. (2023a)). Table 6 presents the comparison results, with further details of these embedders provided in Appendix C.2. The multi-scale periodic embedder consistently outperforms the others across all prediction lengths, while the DFT-based embedder shows declining performance as the prediction length increases. The standard error further indicates that longer forecasting horizons amplify the impact of encoder choice, making it a more critical factor in model accuracy. This highlights the importance of using an unbiased, learnable set encoder to improve generalisation.

**Impact of different number of factors** Previous work, such as Mamba (Gu & Dao, 2023) and xLSTM (Beck et al., 2024), shows that state-space memory size significantly impacts performance. In FACTS, memory size is determined by the number of factors and the dimension of each factor ($d$). To provide a comprehensive analysis, we examine the impact of the preset number of factors ($k$) on FACTS' performance using the MOVi-A videos and the ETTm1 dataset. Specifically, we show the impact of $k$ during testing time in the MOVi-A video reconstruction experiments and the robustness of FACTS in training time in ETTm1 MTS forecasting experiments (with a 96-prediction length setting).

For the MOVi-A experiments, we took a FACTS model that is trained under the "object dicscovery for video reconstruction" setting and evaluated its video reconstruction performance (measured by the image visual quality measure, LPIPS) against different preset $k$ at testing time on the MOVi-A data. Our results in Figure 4 show that the video reconstruction quality can be improved by increasing $k$ up to certain level (> 11). Knowing that the maximum number of objects in these videos (the true causal factors) is 11 (10 objects plus 1 background), suggesting a "sweet point" that is both effective and computationally efficient. However, in practice, identifying such a point in advance can be challenging, Figure 4 indicates that a larger $k$ is preferred.

As the choice of $k$ could largely affect FACTS's performance, we want to investigate how robust FACTS is against different choices of $k$. We examined this on the MTS forecasting task using the ETTm1 data, which consists of 7 variates. To isolate the effect of the number of factors, we train the model across various settings, gridded by different numbers of factors and factor dimensions. As shown in Table 7, FACTS achieves consistent performance across different number of factors (during training) and also the factor dimensions, demonstrating FACTS' robustness to these hyper-parameters.

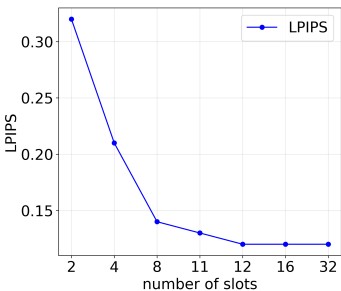

Figure 4: Video reconstruction quality vs number of slots. With video reconstruction quality measure by LPIPS↓.

| Method | ARI ↑ | FG-ARI ↑ | FG-mIoU ↑ |
|---|---|---|---|
| G-SWM (2020) | 57.14 | 49.61 | 24.44 |
| SlotFormer (2023b) | **63.45** | 63.00 | 29.81 |
| SAVi-dyn (2023) | 8.64 | **64.32** | 18.25 |
| **FACTS (Ours)** | 58.25 | 62.34 | **48.11** |

Table 8: Quantitative results for slot dynamics prediction: segmentation quality of predicted future frames, measured by ARI, FG-ARI, and FG-mIoU (all reported in %, higher indicate better).

# E   ADDITIONAL RESULTS

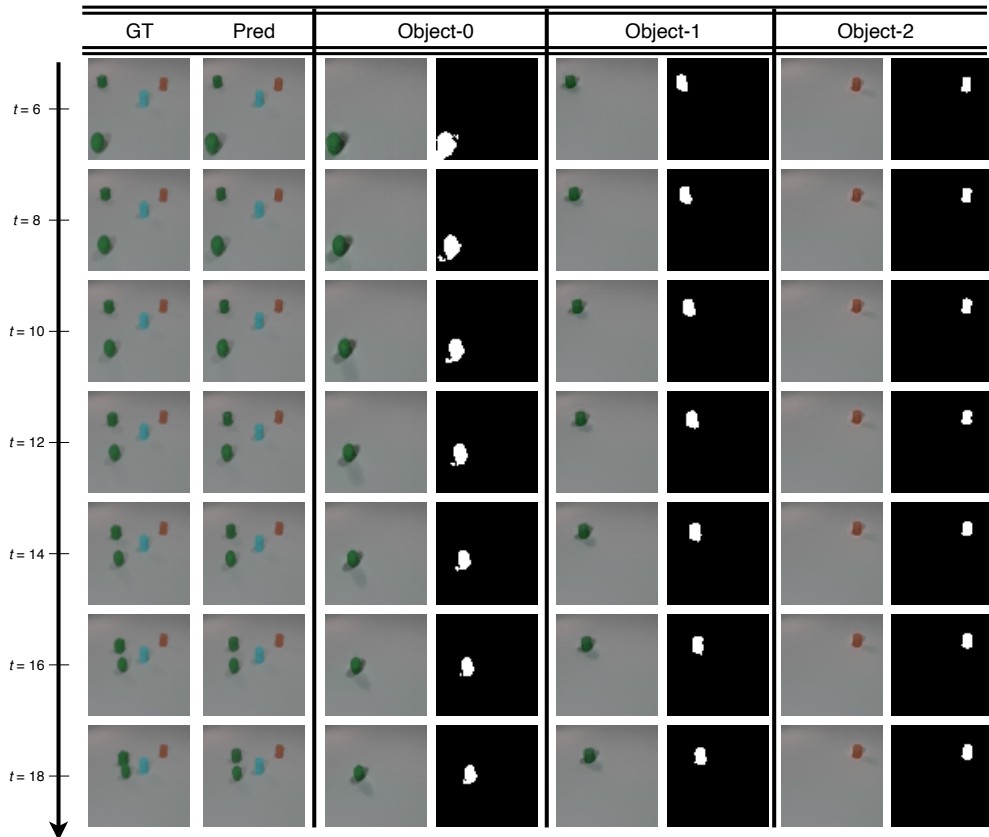

Figure 5: Qualitative results of slot dynamics prediction on CLEVRER. The first two columns show the ground truth (GT) and model predictions (Pred) for future frames. The subsequent columns represent independently rendered dynamics of individual objects (Object-0, Object-1, and Object-2) identified by the model. We show 3 object-centric dynamics in the remaining columns: two columns for each object: the left displays the predicted object dynamics, and the right shows the corresponding object masks.

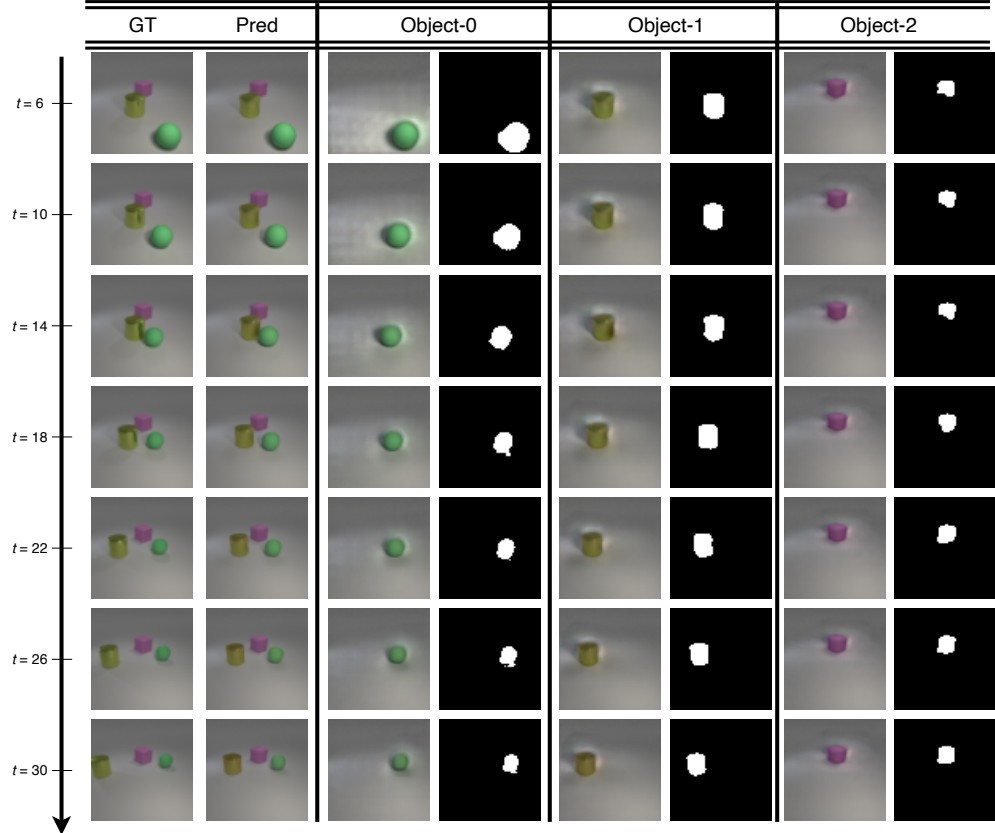

Figure 6: Qualitative results of slot dynamics prediction on OBJ3D. The first two columns show the ground truth (GT) and model predictions (Pred) for future frames. The subsequent columns represent independently rendered dynamics of individual objects (Object-0, Object-1, and Object-2) identified by the model. We show 3 object-centric dynamics in the remaining columns: two columns for each object: the left displays the predicted object dynamics, and the right shows the corresponding object masks.

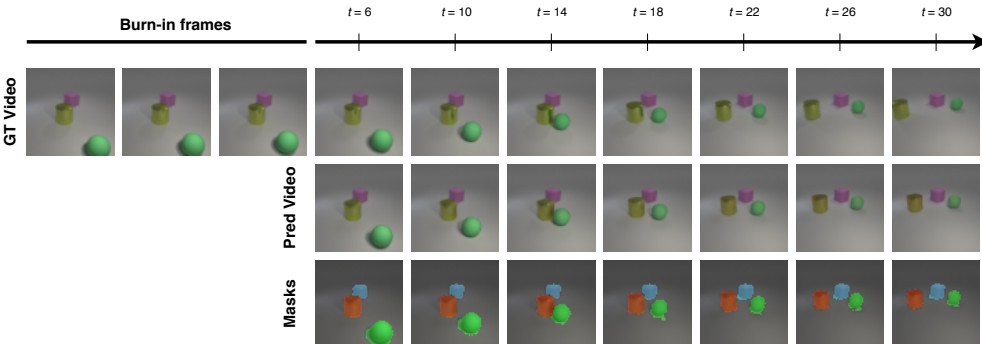

Figure 7: Qualitative results of slot dynamics prediction on OBJ3D. The top row shows the ground truth (GT) video frames, with burn-in frames used for initialisation. The middle row presents the predicted future frames (Pred Video) generated by the model. The bottom row illustrates the object segmentation masks predicted by the model.

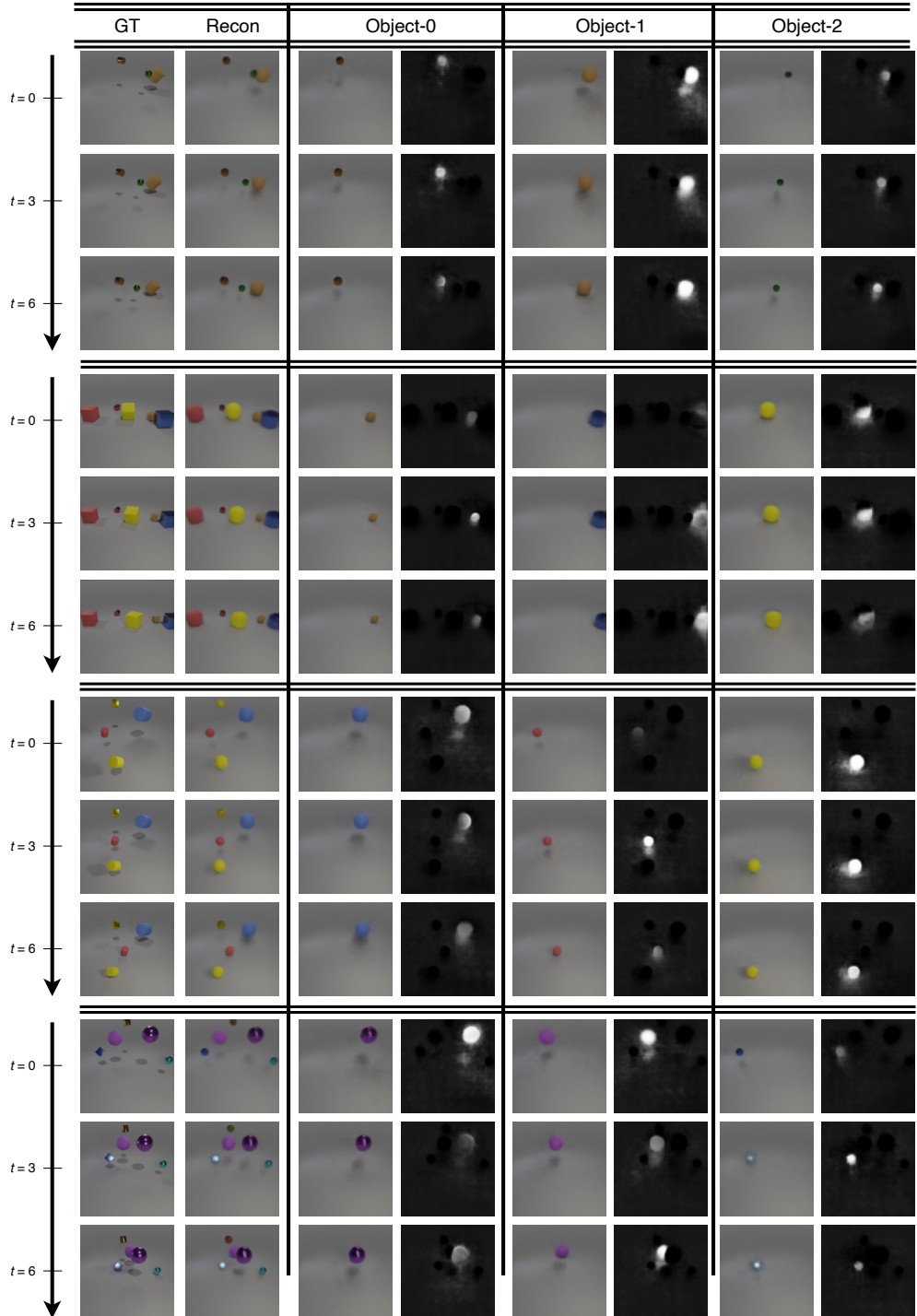

Figure 8: Qualitative results of unsupervised object discovery. The first column shows selected ground truth future frames, while the second column presents our predicted future object-cetric states rendered as frames. We show 3 object-centric dynamics in the remaining columns: two columns for each object: the left displays the predicted object dynamics, and the right shows the corresponding attention masks. FACTS effectively discover and captures interpretable "factors", i.e. objects, for modelling video dynamics.

| Data | Pred.Len. | FACTS (Ours) MSE↓ | MAE↓ | S-Mamba MSE↓ | MAE↓ | iTransformer MSE↓ | MAE↓ | TimesNet MSE↓ | MAE↓ | PatchTST MSE↓ | MAE↓ | DLinear MSE↓ | MAE↓ | Crossformer MSE↓ | MAE↓ | FEDformer MSE↓ | MAE↓ | Autoformer MSE↓ | MAE↓ |
|---|---|---|---|---|---|---|---|---|---|---|---|---|---|---|---|---|---|---|---|
| ETTm1 | 96 | 0.330 | 0.364 | 0.333 | 0.368 | 0.334 | 0.368 | 0.338 | 0.375 | 0.329 | 0.367 | 0.345 | 0.372 | 0.404 | 0.426 | 0.379 | 0.419 | 0.505 | 0.475 |
| | 192 | 0.370 | 0.384 | 0.376 | 0.390 | 0.377 | 0.391 | 0.374 | 0.387 | 0.367 | 0.385 | 0.380 | 0.389 | 0.450 | 0.451 | 0.426 | 0.441 | 0.553 | 0.496 |
| | 336 | 0.400 | 0.404 | 0.408 | 0.413 | 0.426 | 0.420 | 0.410 | 0.411 | 0.399 | 0.410 | 0.413 | 0.413 | 0.532 | 0.515 | 0.445 | 0.459 | 0.621 | 0.537 |
| | 720 | 0.468 | 0.438 | 0.475 | 0.448 | 0.491 | 0.459 | 0.478 | 0.450 | 0.454 | 0.439 | 0.474 | 0.453 | 0.666 | 0.589 | 0.543 | 0.490 | 0.671 | 0.561 |
| | Avg. | 0.392 | 0.397 | 0.398 | 0.405 | 0.407 | 0.410 | 0.400 | 0.406 | 0.387 | 0.400 | 0.403 | 0.407 | 0.513 | 0.496 | 0.448 | 0.452 | 0.588 | 0.617 |
| ETTm2 | 96 | 0.175 | 0.258 | 0.179 | 0.263 | 0.180 | 0.264 | 0.187 | 0.267 | 0.175 | 0.259 | 0.193 | 0.292 | 0.287 | 0.366 | 0.203 | 0.287 | 0.255 | 0.339 |
| | 192 | 0.240 | 0.300 | 0.250 | 0.309 | 0.250 | 0.309 | 0.249 | 0.309 | 0.241 | 0.302 | 0.284 | 0.362 | 0.414 | 0.492 | 0.269 | 0.328 | 0.281 | 0.340 |
| | 336 | 0.304 | 0.341 | 0.312 | 0.349 | 0.311 | 0.348 | 0.321 | 0.351 | 0.305 | 0.343 | 0.369 | 0.427 | 0.597 | 0.542 | 0.325 | 0.366 | 0.339 | 0.372 |
| | 720 | 0.404 | 0.400 | 0.411 | 0.406 | 0.412 | 0.407 | 0.408 | 0.403 | 0.402 | 0.400 | 0.554 | 0.522 | 1.730 | 1.042 | 0.421 | 0.415 | 0.433 | 0.432 |
| | Avg. | 0.281 | 0.325 | 0.288 | 0.332 | 0.288 | 0.332 | 0.291 | 0.333 | 0.281 | 0.326 | 0.350 | 0.401 | 0.757 | 0.610 | 0.305 | 0.349 | 0.327 | 0.371 |
| ETTh1 | 96 | 0.382 | 0.390 | 0.386 | 0.405 | 0.386 | 0.405 | 0.384 | 0.402 | 0.414 | 0.419 | 0.386 | 0.400 | 0.423 | 0.448 | 0.376 | 0.419 | 0.449 | 0.459 |
| | 192 | 0.433 | 0.419 | 0.443 | 0.437 | 0.441 | 0.436 | 0.436 | 0.429 | 0.460 | 0.445 | 0.437 | 0.432 | 0.471 | 0.474 | 0.420 | 0.448 | 0.500 | 0.482 |
| | 336 | 0.474 | 0.440 | 0.489 | 0.468 | 0.487 | 0.458 | 0.491 | 0.469 | 0.501 | 0.466 | 0.481 | 0.459 | 0.570 | 0.546 | 0.459 | 0.465 | 0.521 | 0.496 |
| | 720 | 0.472 | 0.463 | 0.502 | 0.489 | 0.503 | 0.491 | 0.521 | 0.500 | 0.500 | 0.488 | 0.519 | 0.516 | 0.653 | 0.621 | 0.506 | 0.507 | 0.514 | 0.512 |
| | Avg. | 0.440 | 0.428 | 0.455 | 0.450 | 0.454 | 0.447 | 0.458 | 0.450 | 0.469 | 0.454 | 0.456 | 0.452 | 0.529 | 0.522 | 0.440 | 0.460 | 0.496 | 0.487 |
| ETTh2 | 96 | 0.288 | 0.337 | 0.296 | 0.348 | 0.297 | 0.349 | 0.340 | 0.374 | 0.302 | 0.348 | 0.333 | 0.387 | 0.745 | 0.584 | 0.358 | 0.397 | 0.346 | 0.388 |
| | 192 | 0.368 | 0.393 | 0.376 | 0.396 | 0.380 | 0.400 | 0.402 | 0.414 | 0.388 | 0.400 | 0.477 | 0.476 | 0.877 | 0.656 | 0.429 | 0.439 | 0.456 | 0.452 |
| | 336 | 0.414 | 0.427 | 0.424 | 0.431 | 0.428 | 0.432 | 0.452 | 0.452 | 0.426 | 0.433 | 0.594 | 0.541 | 1.043 | 0.731 | 0.496 | 0.487 | 0.482 | 0.486 |
| | 720 | 0.422 | 0.441 | 0.426 | 0.444 | 0.427 | 0.445 | 0.462 | 0.468 | 0.431 | 0.446 | 0.831 | 0.657 | 1.104 | 0.763 | 0.463 | 0.474 | 0.515 | 0.511 |
| | Avg. | 0.373 | 0.399 | 0.381 | 0.405 | 0.383 | 0.407 | 0.414 | 0.427 | 0.387 | 0.407 | 0.559 | 0.515 | 0.942 | 0.684 | 0.437 | 0.449 | 0.450 | 0.459 |
| Electricity | 96 | 0.143 | 0.240 | 0.139 | 0.235 | 0.148 | 0.240 | 0.168 | 0.272 | 0.181 | 0.270 | 0.197 | 0.282 | 0.219 | 0.314 | 0.193 | 0.308 | 0.201 | 0.317 |
| | 192 | 0.155 | 0.252 | 0.159 | 0.255 | 0.162 | 0.253 | 0.184 | 0.289 | 0.188 | 0.274 | 0.196 | 0.285 | 0.231 | 0.322 | 0.201 | 0.315 | 0.222 | 0.334 |
| | 336 | 0.168 | 0.267 | 0.176 | 0.272 | 0.178 | 0.269 | 0.198 | 0.300 | 0.204 | 0.293 | 0.209 | 0.301 | 0.246 | 0.337 | 0.214 | 0.329 | 0.231 | 0.338 |
| | 720 | 0.197 | 0.294 | 0.204 | 0.298 | 0.225 | 0.317 | 0.220 | 0.320 | 0.246 | 0.324 | 0.245 | 0.333 | 0.280 | 0.363 | 0.246 | 0.355 | 0.254 | 0.361 |
| | Avg. | 0.166 | 0.263 | 0.170 | 0.265 | 0.178 | 0.270 | 0.192 | 0.295 | 0.205 | 0.290 | 0.212 | 0.300 | 0.244 | 0.334 | 0.214 | 0.327 | 0.227 | 0.338 |
| Exchange | 96 | 0.081 | 0.197 | 0.086 | 0.207 | 0.086 | 0.206 | 0.107 | 0.234 | 0.088 | 0.205 | 0.088 | 0.218 | 0.256 | 0.367 | 0.148 | 0.278 | 0.197 | 0.323 |
| | 192 | 0.170 | 0.294 | 0.182 | 0.304 | 0.177 | 0.299 | 0.226 | 0.344 | 0.176 | 0.299 | 0.176 | 0.315 | 0.470 | 0.509 | 0.271 | 0.315 | 0.300 | 0.369 |
| | 336 | 0.309 | 0.401 | 0.332 | 0.418 | 0.331 | 0.417 | 0.367 | 0.448 | 0.301 | 0.397 | 0.313 | 0.427 | 1.268 | 0.883 | 0.460 | 0.427 | 0.509 | 0.524 |
| | 720 | 0.808 | 0.677 | 0.867 | 0.703 | 0.847 | 0.691 | 0.964 | 0.746 | 0.901 | 0.714 | 0.839 | 0.695 | 1.767 | 1.068 | 1.195 | 0.695 | 1.447 | 0.941 |
| | Avg. | 0.342 | 0.392 | 0.367 | 0.408 | 0.360 | 0.403 | 0.416 | 0.443 | 0.367 | 0.404 | 0.354 | 0.414 | 0.940 | 0.707 | 0.519 | 0.429 | 0.613 | 0.539 |
| Traffic | 96 | 0.451 | 0.298 | 0.382 | 0.261 | 0.395 | 0.268 | 0.593 | 0.321 | 0.462 | 0.295 | 0.650 | 0.396 | 0.522 | 0.290 | 0.587 | 0.366 | 0.613 | 0.388 |
| | 192 | 0.458 | 0.297 | 0.396 | 0.267 | 0.417 | 0.276 | 0.617 | 0.336 | 0.466 | 0.296 | 0.598 | 0.370 | 0.530 | 0.293 | 0.604 | 0.373 | 0.616 | 0.382 |
| | 336 | 0.472 | 0.302 | 0.417 | 0.276 | 0.433 | 0.283 | 0.629 | 0.336 | 0.482 | 0.304 | 0.605 | 0.373 | 0.558 | 0.305 | 0.621 | 0.383 | 0.622 | 0.337 |
| | 720 | 0.507 | 0.317 | 0.460 | 0.300 | 0.467 | 0.302 | 0.640 | 0.350 | 0.514 | 0.322 | 0.645 | 0.394 | 0.589 | 0.328 | 0.626 | 0.382 | 0.660 | 0.408 |
| | Avg. | 0.472 | 0.303 | 0.414 | 0.276 | 0.428 | 0.282 | 0.620 | 0.336 | 0.481 | 0.304 | 0.625 | 0.383 | 0.550 | 0.304 | 0.610 | 0.376 | 0.628 | 0.379 |
| Weather | 96 | 0.163 | 0.210 | 0.165 | 0.210 | 0.174 | 0.214 | 0.172 | 0.220 | 0.177 | 0.218 | 0.196 | 0.255 | 0.158 | 0.230 | 0.217 | 0.296 | 0.266 | 0.336 |
| | 192 | 0.217 | 0.258 | 0.214 | 0.252 | 0.221 | 0.254 | 0.219 | 0.261 | 0.225 | 0.259 | 0.237 | 0.296 | 0.206 | 0.277 | 0.276 | 0.336 | 0.307 | 0.367 |
| | 336 | 0.275 | 0.299 | 0.274 | 0.297 | 0.278 | 0.296 | 0.280 | 0.306 | 0.278 | 0.297 | 0.283 | 0.335 | 0.272 | 0.335 | 0.339 | 0.380 | 0.359 | 0.395 |
| | 720 | 0.349 | 0.347 | 0.350 | 0.345 | 0.358 | 0.347 | 0.365 | 0.359 | 0.354 | 0.348 | 0.345 | 0.381 | 0.398 | 0.418 | 0.403 | 0.428 | 0.419 | 0.428 |
| | Avg. | 0.251 | 0.278 | 0.251 | 0.276 | 0.258 | 0.278 | 0.259 | 0.287 | 0.259 | 0.281 | 0.265 | 0.317 | 0.259 | 0.315 | 0.309 | 0.360 | 0.338 | 0.382 |
| Solar-Energy | 96 | 0.199 | 0.237 | 0.205 | 0.244 | 0.203 | 0.237 | 0.250 | 0.292 | 0.234 | 0.286 | 0.290 | 0.378 | 0.310 | 0.331 | 0.242 | 0.342 | 0.884 | 0.711 |
| | 192 | 0.249 | 0.271 | 0.237 | 0.270 | 0.233 | 0.261 | 0.296 | 0.318 | 0.267 | 0.310 | 0.320 | 0.398 | 0.734 | 0.725 | 0.285 | 0.380 | 0.834 | 0.692 |
| | 336 | 0.276 | 0.285 | 0.258 | 0.288 | 0.248 | 0.273 | 0.319 | 0.330 | 0.290 | 0.315 | 0.353 | 0.415 | 0.750 | 0.735 | 0.282 | 0.376 | 0.941 | 0.723 |
| | 720 | 0.288 | 0.293 | 0.260 | 0.288 | 0.249 | 0.275 | 0.338 | 0.337 | 0.289 | 0.317 | 0.356 | 0.413 | 0.769 | 0.765 | 0.357 | 0.427 | 0.882 | 0.717 |
| | Avg. | 0.253 | 0.272 | 0.240 | 0.273 | 0.233 | 0.262 | 0.301 | 0.319 | 0.270 | 0.307 | 0.330 | 0.401 | 0.641 | 0.639 | 0.291 | 0.381 | 0.885 | 0.711 |

Table 9: Full results for the MTS long-term forecasting task (in MSE↓ and MAE↓). We compare extensive competitive models under different prediction lengths. The input sequence length is set to 96 for all the datasets above. "Avg" represents the average across all four prediction lengths. For each metric and each dataset, the top performance and the second best are highlighted in **red** and blue, respectively.

