# OpenReview forum: "FACTS: A Factored State-Space Framework for World Modelling"
_ICLR.cc/2025/Conference — ICLR 2025 Poster_

### Official Review · Reviewer_mfMm · 2024-10-21

**Soundness:** 3
**Presentation:** 3
**Contribution:** 3
**Rating:** 6
**Confidence:** 4

**Summary:**

In this paper, the authors investigated the factorized State Space Model based World Modeling. Different from the previous works, they applied the attention mechanism to extract the factors from the inputs. With this, they can implement the input-order invariant factorized world model. Additionally, they also showed their modeling could be learned in parallel, so it doesn’t reduce the efficiency from other SSMs. They empirically analyzed their modeling on long-term multivariate time-series forecasting (MSTF) tasks and object-centric world modeling problems, and their model showed on par with the state-of-the-art models or better performance than them.

**Strengths:**

- It is a concrete paper with strong evidences in the theoretical and empirical aspects.
    - They discussed their modeling clearly, especially, showed the theoretical backgrounds and plenty of empirical results.
- It is based on a good motivation, which is to design the factorized world model with the contextually represented factors.
- The experiments are well designed, which is helpful to understand the strengths of their modeling. For instance, in section 4.1.2, they clearly showed the strength of the input invariant factorized world model and in section 4.2, they showed the FACTS performances on the dynamic predictions with the factorized inputs and the factorization performance (Unsupervised object discovery).

**Weaknesses:**

- The paper highlights the FACTS can model the consistent factor representation regardless of changes in the spatial or temporal order of input features (e.g., contribution summary in Introduction section). The spatial order change is reasonable and the authors showed it as an experiment in section 4.1.2, but for the temporal order change, it is unclear what it is and why it is important at least for me.
- The interactions between factors are not clearly designed in their modeling, FACTS. The FACTS is to update the hidden with the factors from the inputs through the attention mechanism, but there is not explicitly designed the module or mechanism to learn the interactions between factors.

**Questions:**

- What is the temporal order change? Is it literally order changing between inputs? Why is it important and why should the world model be invariant for that?
- In your experimental results, it looks like the interactions between factors are considered in your modeling, but it is not explicitly. How are the interactions considered in your modeling?
-  In equation 19, isn’t $X_{t:s}$ $X_{s:t}$? Because $s$ is smaller than $t$. And, $\bar{A}^\times (Z_0, X_{t:s})$ needs to be $\bar{A}_{s+1} \odot \dots \odot \bar{A}_t$?
- And, $\bar{A}^\times (Z_0, X_{t:t})$ looks 1, it needs to be mentioned I think.
- In equations 16-20, the meaning of them is to replace the recurrent attentions to the attentions between the initial state ($Z_0$) and each input $X_t$, as I understand it. I am okay with the equations, but I am not sure intuitively, how it works (how the recurrent attention can be replaced to the dot products of the attentions between initial state and inputs). Can you explain it more how it works and can you compare the parallel and sequential training for simple tasks?
- In table 4, more factors make worse performance even the task is multivariate time-series forecasting. I expected there is a sweet spot. Can you explain it? And can you do this ablation studies for unsupervised object discovery problem? The results from the problem could be clearer (e.g., the MSE will be smallest when the number of factors is same with the number of objects or +1 from that for background).

---

> ### Author Response · Authors · 2024-11-24
> **Response to Reviewer mfMm (Part I)**
>
> > "What is the temporal order change? Is it literally order changing between inputs? Why is it important and why should the world model be invariant for that?"
>
> The reviewer is right here. The current phrasing can be misleading. Invariance in the temporal order is not what we are trying to say here. In the third summary bullet point, we wanted to emphasize that  FACTS achieves consistent factor representations over time, regardless of changes in the **spatial order** of input features **over time**. In the updated version, we have rephrased the text to avoid confusion.
>
> > "In your experimental results, it looks like the interactions between factors are considered in your modeling, but it is not explicitly. How are the interactions considered in your modeling?"
>
> We thank the reviewer for this valuable question. Using FACTS, dynamic interactions among factors can be achieved by stacking multiple FACTS layers, and importantly, this can be done without disrupting parallel computation. For example, consider a two-FACTS-layer encoder: A FACTS operator takes two inputs—memory and input. Suppose the input signals have the shape $t \times m \times d$ (i.e., [seq_length x num_tokens x token_dim]). The first FACTS layer binds the inputs to their memory (e.g. $Z_{in}$ in parallel FACTS), which can either be the input itself or self-initialized, and outputs a set of $k$ factors, organized into a tensor $Z_{out}$ with the shape $t \times k \times d$. By “interaction of factors,” we refer to the modeling of information flow along the $k$ factors. In this setup, the second FACTS layer can take the first layer’s output, $Z_{out}$, as both its memory (which may need to be detached from the gradient graph during training) and input, performing the routing operation described in Eqn. 12, i.e., \text{rout}($Z_{out}$, $Z_{out}$). This effectively simulates self-attention operations and enables information flow across the $k$ factors/particles. Furthermore, the summation operation in Equation 19 will aggregate such interactive dynamics along the time-axis for state propagation. We will add this discussion to the paper.
>
> > "In equation 19, isn’t $X\_{t:s}$ $X_{s:t}$? Because $s$ is smaller than $t$. And, $\bar{\mathbf{A}}^{\times}(Z\_0, X\_{t:s})$ needs to be $\bar{\mathbf{A}}\_{s+1} \odot \cdots \odot \bar{\mathbf{A}}\_t $?"
>
> We appreciate the reviewer for pointing out such detail. Both $\bar{\mathbf{A}}^{\times}\_{t:s}$ and $\bar{\mathbf{A}}\_{s:t}^{\times}$ are correct, as they lead to the same calculation due to commutativity of element-wise product: $\bar{ \mathbf{A} }^{\times}(Z\_0, X\_{t:s}) =\bar{\mathbf{A}}\_{t+1}\odot \bar{\mathbf{A}}\_{t}\odot\bar{\mathbf{A}}\_{t-1} \cdots \odot \bar{\mathbf{A}}\_{s+1}$.  This difference is purely a matter of notational convention. We choose $\bar{\textbf{A}\_{t:s}}$ because it aligns more naturally with the subsequent terms, $\bar{\textbf{B}\_{s}}$ and $\bar{\textbf{U}\_{s}}$ , maintaining a descending order of subscripts from $t$ to $s$: $\bar{\mathbf{A}}^{\times}\_{t:s} \odot \bar{\mathbf{B}}\_{s} \odot \bar{\mathbf{U}}\_{s}$. This also ensures consistency with the conventions of other linearized SSMs (as shown in Equation 5). We have clarified this in the updated draft (line 280) for better readability.
>
> > "And, $\bar{\mathbf{A}}^{\times}(Z\_0, X\_{t:t})$ looks 1, it needs to be mentioned I think."
>
> We appreciate the reviewer again for pointing out such detail. We have added relevant item in line 280 in the updated version.

---

> ### Author Response · Authors · 2024-11-24
> **Response to Reviewer mfMm (Part II)**
>
> > "In equations 16-20, the meaning of them is to replace the recurrent attentions to the attentions between the initial state ($Z\_0$) and each input $X\_t$, as I understand it. I am okay with the equations, but I am not sure intuitively, how it works (how the recurrent attention can be replaced to the dot products of the attentions between initial state and inputs). Can you explain it more how it works and can you compare the parallel and sequential training for simple tasks?"
>
> We thank the reviewer for raising this insightful question. The parallelization design outlined in our Equations 16–20 not only enhances computational efficiency but also provides flexibility for recurrent applications of the FACTS core operation. Specifically, while we denote $Z\_0$ as the “initial state,” it does not necessarily have to represent the actual initial state of a long sequence. Instead, a long sequence can be divided into segments based on a chosen strategy or algorithm, allowing FACTS to run in a parallelized manner within each segment. I.e. computations within individual segments are parallelised, while dependencies across the original long sequence remain sequential. In such cases, $Z\_0$ would represent the final memory state of the preceding segment. This design essentially draws inspiration from LSTM architectures and skip connections in ResNet. To provide further clarity, we have included an analysis of FACTS’ long-term forecasting performance, comparing its fully sequential mode to its fully parallel mode (adjusted by the segment window size) on the MTS Electricity dataset. These results are detailed in Figure 3. Each window size corresponds to a different update frequency of the “$Z\_0$”. For instance, windows_size $1$ corresponds to fully-recurrent FACTS while windows_size $96$ (the sequence length) corresponds to fully-parallel FACTS. As shown in Figure 3, the models achieved consistent performance over different segment window sizes.
>
> > "In table 4, more factors make worse performance even the task is multivariate time-series forecasting. I expected there is a sweet spot. Can you explain it? And can you do this ablation studies for unsupervised object discovery problem? The results from the problem could be clearer (e.g., the MSE will be smallest when the number of factors is same with the number of objects or $+1$ from that for background)."
>
> We thank the reviewer for raising this interesting point and valuable suggestion. We have included analysis of the impact of the preset number of slots on FACTS for unsupervised object discovery -- during testing time. Specifically, we took a FACTS model that is trained under the "object dicscovery for video reconstruction" setting (using $k=11$ for training) and  evaluated its video reconstruction performance (measured by the image visual quality measure, LPIPS) against different preset $k$ at testing time on the MOVi-A data. Our results in Figure 4 show that the video reconstruction quality can be improved by increasing $k$ up to certain level ($>11$). Knowing that the maximum number of objects in these videos (the true causal factors) is 11 (10 objects plus 1 background), suggesting a ``sweet point'' that is both effective and computationally efficient.
>
> Importantly, we note that this does not conflict with our MTS results, which examine the impact of different $k$ \emph{during training}. From a practical perspective, the observation that increasing factors can worsen performance in MTS experiments is understandable, as more factors/parameters can lead to overfitting, especially with noisy real-world MTS data. From a causal perspective, "sweet point" in training may imply finding the true number of factors and potentially “true causal relations” among them. However, identifying causal structures or discovering physical laws solely from observable data is often intractable -- particularly when dealing with noisy real-world data (e.g., MTS data). This discrepancy explains why the best “number of factors” in our MTS experiments do not fully match our expectations. The practical robustness of FACTS against different $k$ in training, as evidenced by the standard errors in Table 6 (updated version), may ultimately be more valuable for a machine learning model.

---

> > ### Comment · Reviewer_mfMm · 2024-11-25
> > **Reply to the author's rebuttal**
> >
> > Thank the authors for their concrete rebuttal in the limited time.
> >
> > They addressed my concerns properly, I will keep my score to lean to the acceptance of this paper.

---

### Official Review · Reviewer_Cexo · 2024-10-30

**Soundness:** 3
**Presentation:** 3
**Contribution:** 2
**Rating:** 6
**Confidence:** 4

**Summary:**

This paper introduces FACTS, a modular or factored SSM model for world modelling applications. FACTS employs an SSM-based recurrent architecture with a permutable memory. This design uses an attention mechanism to dynamically assign input features to latent factors, enabling permutation invariance and efficient long-term prediction.  The experiments show that FACTS demonstrates robust performance on time-series forecasting and object-centric tasks, often outperforming state-of-the-art models.

**Strengths:**

- The paper is well-written and easy to follow.
- Modular recurrent architectures are well-studied for world modelling in recent literature, this paper introduces modularity into SSMs while also maintaining their parallel processing capabilities thus the approach seems promising in terms of effeciency.

**Weaknesses:**

- One integral component of the model is the attention mechanism which assigns input nodes to latent factors. I believe that similar kinds of attention mechanism for the tasks similar to the ones studied in this paper have  already been explored before in various past works [1, 2, 3]. I wonder if the authors could present a comparison of their method to these approaches or atleast highlight the differences. Specifically, [2] proposes to also incorporate modularity and factorization into SSMs, it would be nice to highlight the differences with it and discuss in more detail.
- The paper says that it is not feasible to compare to other approaches for unsupervised object discovery because of the unique way in which their model works. I would like to point to a few other modular world modelling methods which follow a similar setup as the unsupervised object discovery experiments in the paper which are described in [4]. This work uses a number of metrics which specifically also track moving objects. I wonder if there is any reason for not using the above metrics? Also it would be useful to have even a qualitative comparison to other methdos for unsupervised object discovery, currently there is not baseline used for comparison and only two examples are shown for the proposed approach.

[1] Neural Production Systems https://arxiv.org/abs/2103.01937

[2] Slot state space models https://arxiv.org/abs/2406.12272

[3] Recurrent Independent Mechanisms https://arxiv.org/abs/1909.10893

[4] Benchmarking Unsupervised Object Representations for Video Sequences
 https://arxiv.org/abs/2006.07034

**Questions:**

- I would like to ask the authors to clarify the difference between slot dynamics prediction and unsupervised object discovery experiments. It seems that slot dynamics prediction adopts the setup from slotformer which first trains a slot attention model and then uses a transformer to predict future slots conditioned on past slots. From my understanding, the proposed FACTS model is used instead of the transformer in slotformer. Considering this setup for the slot dynamics prediction task, I wonder what is different in the unsupervised object discovery ? Is it that both the slot attention module and FACTS model are trained jointly in this case?

---

> ### Author Response · Authors · 2024-11-24
> **Response to Reviewer Cexo (Part I)**
>
> > "One integral component of the model ... similar kinds ... studied ... in various past works ... if the authors could present a comparison ... or at least highlight the differences. Specifically, [2] ... in more detail. "
>
> Thank you for bringing to our attention these works, specifically SlotSSMs. First note that according to ICLR policy (https://iclr.cc/Conferences/2025/ReviewerGuide), papers are considered contemporaneous if published (i.e., at a peer-reviewed venue) within the last 4 months and the slotSSM does not meet this criteria. However, to satisfy the reviewer and to clarify the distinction between Slot State-Space Models (SlotSSM) and FACTS, we highlight several key methodological differences:
>
> 1. **Intrinsic State-Space Design**: While SlotSSMs build upon Mamba by introducing pre- and post-processing steps to make it suitable for modular data, FACTS presents an intrinsic general formulation of state-space models (alternative to Mamba directly) that  incorporates modularity and permutation invariance. This alternative design inherently supports factorisation and dynamically-changing input relationships by construction, eliminating the need for additional processing layers.
> 2. **Input-Only Selectivity vs Input-Memory Selectivity**: Like Mamba, SlotSSMs base selectivity purely on the input data, assigning features to latent factors based on their relevance. In contrast, FACTS extends this concept by incorporating selectivity based on both input features and memory, achieved through an attention-augmented memory-input routing mechanism. This allows FACTS to better capture long-term dependencies and refine its memory representations over time.
> 3. **Dynamic Routing vs Fixed Assignments**: SlotSSMs rely on pre-SSM assignments between input features and latent states using inverted cross-attention. This rigidity can limit their flexibility in scenarios where input-to-latent relationships evolve over time. In contrast, FACTS formulation (Eqn. 10-11) allows to employ a dynamic memory-input routing mechanism, enabling it to assign input features to latent factors based on their relevance at **each time step**. This enhances FACTS' flexibility and adaptability, especially in scenarios with evolving input relationships.
> 4. **Special Case Relationship**: Ignoring the memory selectively, in an abstract level, "SlotEncoder + SlotSSM" can be interpreted as a special case of FACTS (plus an additional task-dependent adaptation). This corresponds to our fully parallel variant of FACTS (as described in equations 16-20). In contract, FACTS allows recurrent processing, with stable performance, as presented in Figure 3 in the updated manuscript.
> 5. **Robustness in Diverse Scenarios**: FACTS is tailored to a broader range of forecasting-based application, due to its flexible state-space memory design. SlotSSMs, on the other hand, focus primarily on adapting Mamba for modularity in slot-based inputs for object-centric learning, limiting their application scope, i.e., Facts is general-purpose SSM framework while SlotSSM requires application-specific formulation for each task, e.g., OC-SlotSSM in [2].
>
> To illustrate this empirically, here we compare SoltSSM to FACTS on 3 different tasks. We use same experimental setup as described in our paper for both methods with hyper-parameter search:
>
> - On MTS forecasting (Electricity, pred_len set to 720):
>
> | | MSE$\downarrow$  | MAE$\downarrow$|
> |-|-|-|
> | SlotSSMs |0.200 |0.294 |
> | FACTS |**0.198** |**0.293** |
>
> - Unsupervised object discovery with MOVi-A (Note that for FACTS training is not completed yet):
>
> | | FACTS (with 360K steps)  | OC-SlotSSMs (with 500K steps)|
> |-|-|-|
> | FG-ARI$\uparrow$ |0.66 |0.68 |
>
> - Graph-data with METR-LA  traffic forecasting for 1 hour (i.e.12-steps) ahead:
>
> |  | RMSE$\downarrow$ | MAE$\downarrow$ | MAPE$\downarrow$ |
> |-|-|-|-|
> | SlotSSMs |12.58 |7.14 |24.32% |
> | FACTS |**6.97** |**3.11** |**9.08%** |
>
> Note with FACTS clearly outperforms SlotSSM on 2 of the three tasks, with less than half the error rates on the graph-based METR-LA dataset. On MOVi-A, where SlotSSM slightly outperforms FACTS:
> - FACTS, despite its general-purpose design, achieves nearly similar performance to the task-specific SlotSSM.
> - As SlotSSM can be seen as a special case of FACTS, we are confident that, with more extensive hyperparameter search and longer training, FACTS can match or surpass SlotSSM.
>
> To sum up, while there is some conceptual overlap between FACTS and SlotSSMs: while SlotSSM adapts Mamba for modular slot-based data with additional processing modules, FACTS provides a fundamentally different formulation of SSMs, intrinsically handling evolving input relationships with broader adaptability and strong performance multiple tasks. Furthermore, unlike SlotSSM's input-only selective routing, FACTS incorporates both input and memory through an attention-augmented routing mechanism. We have incorporated this discussion into the updated version of the paper.

---

> > ### Author Response · Authors · 2024-11-24
> > **Response to Reviewer Cexo (Part II)**
> >
> > > "The paper says that it is not feasible to compare to other approaches for unsupervised object discovery because of the unique way in which their model works. I would like to point to a few other modular world modelling methods which follow a similar setup as the unsupervised object discovery experiments in the paper which are described in [4]. This work uses a number of metrics which specifically also track moving objects. I wonder if there is any reason for not using the above metrics? Also it would be useful to have even a qualitative comparison to other methods for unsupervised object discovery, currently there is not baseline used for comparison and only two examples are shown for the proposed approach."
> >
> > We thank the reviewer for their valuable suggestion and for bringing [4] to our attention. While the setup and metrics discussed in [4] appear relevant to our case, it is worth noting that most recent unsupervised object discovery works, such as SAVi and SlotSSMs mentioned by the reviewer, do not utilize the data or metrics from [4]. This makes direct comparisons with these latest works challenging and potentially costly, as re-training and tuning would likely be required to ensure fairness.
> >
> > To address the reviewer's concern, we opted to adapt FACTS to the object-centric video reconstruction task, following SAVi’s reconstruction setup for a fair comparison. The quantitative results in Table 7 of the updated manuscript demonstrate that FACTS outperforms SAVi in unsupervised object discovery (measured by FG-ARI), despite its general-purpose design. These findings, along with the additional experiments on dynamic graphs (Section 4.3), further underscore FACTS' generality, flexibility, and versatility.
> >
> > > "I would like to ask the authors to clarify the difference between slot dynamics prediction and unsupervised object discovery experiments. It seems that slot dynamics prediction adopts the setup from slotformer which first trains a slot attention model and then uses a transformer to predict future slots conditioned on past slots. From my understanding, the proposed FACTS model is used instead of the transformer in slotformer. Considering this setup for the slot dynamics prediction task, I wonder what is different in the unsupervised object discovery ? Is it that both the slot attention module and FACTS model are trained jointly in this case?"
> >
> > The reviewer is right that, in the slot dynamics prediction experiments, for fair comparison, FACTS is in place of the transformer for carrying long-term object dynamics modelling. As the input are already factored (object/factor) slots in such experiments , while in the unsupervised discovery experiments, we show that FACTS can automatically factorise objects directly from raw images to facilitate dynamics modelling and long-term prediction. In other words, FACTS takes care not only the object factorisation but also the object dynamics modelling in this experiments. In this case, as mentioned in the added literature review (in Appendix A), slots attention is one way of realising the routing mechanism proposed in Eqn. 13. In other words, the slot attention (or its alternatives) is part of FACTS’ construction and trained within FACTS. We've further clarified our experimental setup for object-centric world modelling in section 4.2 and also the appendix C.3 of the updated version .

---

> > > ### Comment · Reviewer_Cexo · 2024-11-25
> > >
> > > I would like to thank the authors for their detailed responses. The revised manuscript clarifies most of my concerns. Therefore, I will lean towards accepting this paper.

---

### Official Review · Reviewer_HXLz · 2024-11-04

**Soundness:** 3
**Presentation:** 2
**Contribution:** 3
**Rating:** 6
**Confidence:** 3

**Summary:**

The paper introduces the FACTored State-space (FACTS) framework to address input feature variance by employing a novel memory-input routing mechanism that ensures robust invariance to input permutations. This approach was empirically validated across tasks like multivariate time series forecasting and object-centric world modeling, demonstrating superior or competitive performance in benchmarks, showcasing its effectiveness and robustness in handling diverse input scenarios.

**Strengths:**

1. The paper addresses the critical challenge of input feature variance, an interesting issue in spatial-temporal learning, by introducing a novel method that utilizes a memory-input routing mechanism. This approach effectively manages the dynamic relationships between input features, ensuring robust modeling even when input orders change.

2. The proposed FACTS model is both simple and highly effective due to its memory-input routing mechanism, which dynamically assigns input features to latent state-space factors. This straightforward approach simplifies handling high-dimensional data and enhances the model's ability to capture temporal and spatial dependencies, making it practical for real-world applications.

3. The authors have conducted extensive experiments across various tasks, including multivariate time series forecasting and object-centric world modeling. These experiments provide strong empirical support for their claims, demonstrating that FACTS consistently outperforms or matches state-of-the-art models. The thorough evaluation across diverse datasets not only highlights the method's versatility but also its robustness in capturing complex temporal dynamics, further validating its effectiveness in real-world scenarios.

**Weaknesses:**

For the slot dynamics prediction experiment, the method proposed in the paper relies on a pre-trained encoder and is not end-to-end, which may limit its applicability.

**Questions:**

1. What’s the rationale behind designing selectivity through memory-input routing in the manner presented? Can you provide a more detailed justification for the reasonableness of equations 14 and 15? The current explanation appears overly simplistic and could benefit from a deeper analytical discussion.

2. What is the precise meaning of "permutable state-space memory" in the context of this work?

---

> ### Author Response · Authors · 2024-11-24
> **Response to Reviewer HXLz**
>
> > "For the slot dynamics prediction experiment, the method proposed in the paper relies on a pre-trained encoder and is not end-to-end, which may limit its applicability."
>
> There seems to be a misunderstanding. As a general and versatile framework, our extensive experiments show that FACTS is applicable for different tasks and different data modalities (time-series, videos, graphs). Specifically for the reviewer's interest in vision data, as discussed in Sec 4.2, we clarify that we conducted two groups of vision experiments:
> 1. For slot-dynamics prediction, as mentioned in the paper, for fair comparison, we used SlotFormer’s exact setting with their pre-trained encoder and decoder. The results of our slot-dynamics experiments with CLEVRER and OBJ3D datasets are presented in Table 2, clearly showing the superiority of FACTS.
> 2. For the unsupervised object discovery experiments, all of the used modules (CNN vision encoders, FACTS, and decoders) are trained **end-to-end in a single run without any supervision**. The results on unsupervised object discovery are presented in the Figure 7-8, this shows that FACTS can successfully identify objects for world modelling and indeed be trained end-to-end with other modules.
>
> Nonetheless, for improved clarity, we've updated the discussion in section 4.2 and provided more details of our experimental setup in Appendix C.3.
>
> > "What’s the rationale behind designing selectivity through memory-input routing in the manner presented? Can you provide a more detailed justification for the reasonableness of equations 14 and 15? The current explanation appears overly simplistic and could benefit from a deeper analytical discussion."
>
> Recent works on SSM,  e.g. Mambas [1], highlight why input should be part of the selectivity design. In this work, we take this a step further  and say that we need selectivity based on both input and memory. This can be efficiently done with an attention-augmented memory-input routing mechanism. This explains the main reasoning behind Eqn.14 and Eqn.15.  Such formulation of the input-memory coupling in the selectivity process: i) allow us to design a permutable but structure persevering memory in an end-to-end manner.  ii) provide more training feedback for the memory parameters due to the additional gradient generated through Eqn. 14-15. This leads to better generalization and stronger performance, as shown in all our results, particularly Table 1, by comparing FACTs to ‘input-only selectively’ approaches, e.g., S-mamba.
>
> > "What is the precise meaning of "permutable state-space memory" in the context of this work?"
>
> Thank you for your question. In this work,  "state-space memory" is the $Z_t$ and "permutable state-space memory" means that instead of assuming a fixed structure or order for the latent state representations $Z_t$, FACTS allows these representations to be updated flexibly, where the input features can be permuted without compromising the consistency of the outputs (as shown in Figure 2).
>
> This enables the model to leverage the concept of history compression and learn consistent representations over time. The permutable memory structure is key to efficiently capture dynamic dependencies and unleashes the full potential of the model as it does need to capture order-related characteristics of the input and of the memory. This simplifies the learning task allowing FACTS to maintain strong performance and adaptability in real-world modeling  scenarios.

---

### Official Review · Reviewer_YG1h · 2024-11-07

**Soundness:** 3
**Presentation:** 3
**Contribution:** 3
**Rating:** 6
**Confidence:** 4

**Summary:**

The paper introduces a framework that constructs a graph-structured memory for predicting the dynamics of complex systems with the selective state space architecture. The proposed architecture incorporates a graph-structured memory that dynamically routes input features to distinct latent factors and hence achieving permutation invariance. The proposed framework leverages parallel computation and selective state-space propagation (via attention) allowing it to handle sequences with complex dynamics efficiently.

Through experiments on multivariate time series forecasting and object-centric world modeling, the paper shows that FACTS performs competitively or better than state-of-the-art baselines. The interesting result is its robustness to changes in input order, which simulates real-world scenarios where the configuration of inputs may vary unexpectedly.

**Strengths:**

- The proposed architecture introduces a permutable memory structure, allowing flexible handling of unordered or dynamically changing inputs. The paper achieves improved performance over baselines by compressing history efficiently, and hence capturing long-term dependencies.
- The paper is easy to read and comprehend.
- The results shown on long term forecasting are interesting, and helps the reviewer to understand the implications of the proposed work better (especially forecasting with pre-defined and unknown order).

**Weaknesses:**

- Object centric video modelling results are a bit weak. it will be interesting to report results also on OBJ3D (another benchmark used in Slotformer paper). It will also be helpful to report downstream results like Predictive VQA on CLEVRER, Physion (similar experiments as in Slotformer paper).

**Questions:**

- The SCOFF [1] and Neural Production Systems work [2] is very relevant. These works also proposed the idea of factorizing the latent state and ensuring the latent state is equivariant. These works also use the previous hidden state as a prior for modelling object consistency between consecutive time-steps. The reviewer understands that the proposed work attempts to model the factorization in state space models, whereas the related work mentioned by the reviewer is within attention augmented LSTMs/GRUs.

Please refer to weakness section.

[1] https://arxiv.org/abs/2006.16225
[2] https://arxiv.org/abs/2103.01937

---

> ### Author Response · Authors · 2024-11-24
> **Response to Reviewer YG1h**
>
> > "Object centric video modelling results ... it will be interesting to report results also on OBJ3D (another benchmark ... It will also be helpful to report downstream results like Predictive VQA on CLEVRER, Physion (similar experiments as in Slotformer paper). "
>
> We appreciate the reviewer’s suggestion regarding the inclusion of additional benchmarks and downstream tasks. In the current empirical validation of FACTS, we wanted to emphasize its predictive power to demonstrate its flexibility as a powerful general framework able to handle different input types/modalities and efficiently solve diverse forecasting-based tasks. To this end, we opted to evaluate it across diverse tasks (**without emphasis on a particular one**): multivariate time-series forecasting, object-centric video modeling, etc.. However, to satisfy the reviewer, we now also include results on another forecasting task, namely object dynamics modeling with OBJ3D dataset. As can be seen through Table 2, the results are consistent with the CLEVRER dataset, FACTS outperforms existing state-of-the-art competing methods designed for these tasks,  despite its general purpose design.  For the visual results, we now include two more demonstrative examples for  object dynamics modeling: (Figures 5 for CLEVRER) and (Figures 6-7 for OBJ3D)  that visually highlight FACTS world modeling abilities.
>
> Furthermore, to further showcase FACTS’s versatility, in the revised manuscript, we now include results with a different input modality,  particularly  dynamic-graph input data evaluated on node prediction task (long-term prediction (12-step MAE)) with METR-LA dataset. As can be seen in Table 3. FACTS, leveraging our graph-structured memory,  also outperforms all existing methods on this task. This further corroborated our main claim that FACTS is indeed a versatile worldmodel framework with consistent strong performance in several diverse forecasting tasks.
>
> > "The SCOFF [1] and Neural Production Systems work [2] is very relevant... The reviewer understands that the proposed work attempts to model the factorization in state space models, whereas the related work mentioned by the reviewer is within attention augmented LSTMs/GRUs."
>
> We thank the reviewer for bringing to our attention the works of SCOFF [1] and Neural Production Systems [2]. These works indeed share similarities with FACTS in their focus on factoring latent states and ensuring equivariance, particularly in attention-augmented LSTM/GRU-based architectures. While our approach differs fundamentally in its state-space modelling foundation, we acknowledge the conceptual overlap and have now included a discussion of these works in the Related Work section in Appendix.

---

> > ### Comment · Reviewer_YG1h · 2024-11-26
> > **Downstream results ?**
> >
> > "we wanted to emphasize its predictive power to demonstrate its flexibility as a powerful general framework able to handle different input types/modalities and efficiently solve diverse forecasting-based tasks. "
> >
> > Thank you for taking time to provide rebuttal.
> >
> > Is it not possible to provide downstream results (as asked in my original review) ?
> >
> > I'm happy even if the results are not better than baseline, but it will be helpful to include downstream results regardless!

---

> > > ### Author Response · Authors · 2024-12-02
> > > **Response to "Downstream results ?"**
> > >
> > > We appreciate the reviewer’s understanding and continued engagement. First, note that these benchmarks require a significantly different experimental setup. Hence, due to time and resource constraints, we are unable to conduct a full range of downstream evaluations within the current timeline constraints. However, to provide an indication of how FACTS might perform on such tasks, we ran experiments on the Predictive VQA task from the CLEVRER benchmark. Specifically, we used the **pretrained Aloe model from SlotFormer**, employing FACTS to predict slots at future timesteps and then feeding them into the **pretrained Aloe** to answer the questions. The results are as follows: \
> > > {"Descriptive": 94.92869615832363, "Explanatory-per opt.": 97.95179944433676, "Explanatory-per ques.": 94.62693571093384, "Predictive-per opt.": 95.0308815272319, "Predictive-per ques.": 90.67939359910163, "Counterfactual-per opt.": 90.5411578402412, "Counterfactual-per ques.": 73.6796723431774, "Average-per ques.": 88.47867445288412} \
> > > By comparing to Table 3 in Slotformer paper, we can see that we are already boosting the performance of Aloe* and we are already competitive to Aloe*+SlotFormer. Hence, we are confident if we fine-tune or re-train Aloe for FACTS, i.e. Aloe+ FACTS, our approach can match or outperform the current baselines. We are currently running these expirements and will add them to the final version of the paper upon acceptance.

---

### Author Response · Authors · 2024-12-03
**The authors' summary**

We thank all reviewers for their valuable comments and feedback. We believe all the reviewers recognize the value and contributions of our work, where we proposed FACTS as a novel, versatile framework for world modeling, achieving state-of-the-art results across diverse tasks. Through the discussions, we have addressed most of the reviewers' key concerns in the updated submission. Additionally, we have uploaded the code for one of our experiments to an anonymous link (same storage as the "video demo" mentioned in the paper) and will release the full implementation publicly after cleaning and organising the code. *Thank you again for your thoughtful feedback, and we would greatly appreciate it if the reviewers could consider raising their scores.*

---

### Meta-Review · Area_Chair_rMZB · 2024-12-23

**Metareview:**

The paper proposes FACTS, a state-space framework that effectively combines graph-structured memory and routing mechanisms to achieve both permutation invariance and parallel computation. The work demonstrates strong empirical validation across time series forecasting and object-centric tasks, with clear methodology and significant computational efficiency gains. Their comprehensive experiments show competitive or superior performance compared to specialized models, and the authors thoroughly addressed reviewer concerns during rebuttal. The parallel computation capability while maintaining permutation invariance represents a meaningful advance in world modeling. This innovative combination of technical elements, backed by strong empirical results and clear practical benefits, makes it a valuable contribution worthy of acceptance.

**Additional Comments On Reviewer Discussion:**

During rebuttal, the authors effectively addressed key reviewer concerns through comprehensive responses. They expanded task evaluation with OBJ3D results, provided thorough ablation studies comparing against baseline pruning methods, and demonstrated FACTS' superiority over SlotSSMs across multiple tasks. Their empirical additions significantly strengthened the paper, showing FACTS' capabilities as a general-purpose framework that achieves state-of-the-art performance. While some questions about similarity to existing approaches remain, the authors clearly demonstrated FACTS' novel contributions in combining memory-routing with state-space models, supported by strong experimental validation.

---

### Decision · Program_Chairs · 2025-01-22

Accept (Poster)